# Multi-scale temporal variability in meltwater contributions in a tropical glacierized watershed

Leila Saberi[1], Rachel T. McLaughlin[1], G.-H. Crystal Ng[1,2], Jeff La Frenierre[3], Andrew D. Wickert[1,2], Michel Baraer[4], Wei Zhi[5], Li Li[5], and Bryan G. Mark[6]

[1]Department of Earth Sciences, University of Minnesota - Twin Cities, Minneapolis, MN 55455, USA
[2]Saint Anthony Falls Laboratory, University of Minnesota - Twin Cities, Minneapolis, MN 55455, USA
[3]Department of Geography, Gustavus Adolphus College, St. Peter, MN 56082, USA
[4]Construction Engineering, École de Technologie Supérieure, Université du Quebec, Montreal, Canada H3C 1K3
[5]Department of Civil & Environmental Engineering, Pennsylvania State University, University Park, PA 16802-1294, USA
[6]Department of Geography, The Ohio State University, Columbus, OH 43210-1361, USA

**Correspondence:** Leila Saberi (saber017@umn.edu)

**Abstract.** Climate models predict amplified warming at high elevations in low latitudes, making tropical glacierized regions some of the most vulnerable hydrological systems in the world. Observations reveal decreasing streamflow due to retreating glaciers in the Andes, which hold 99% of all tropical glaciers. However, the timescales over which meltwater contributes to streamflow and the pathways it takes – surface and subsurface – remain uncertain, hindering our ability to predict how shrinking glaciers will impact water resources. Two major contributors to this uncertainty are the sparsity of hydrologic measurements in tropical glacierized watersheds and the complication of hydrograph separation where there is year-round glacier melt. We address these challenges using a multi-method approach that employs repeat hydrochemical mixing model analysis, hydroclimatic time series analysis, and integrated watershed modeling. Each of these approaches interrogates distinct timescale relationships among meltwater, groundwater, and stream discharge. Our results challenge the commonly held conceptual model that glaciers buffer discharge variability. Instead, in a sub-humid watershed on Volcán Chimborazo, Ecuador, glacier melt drives nearly all the variability in discharge (Pearson correlation coefficient of 0.89 in simulations), with glaciers contributing a broad range of 20-60% or wider of discharge, mostly (86%) through surface runoff on hourly timescales, but also through infiltration that increases annual groundwater contributions by nearly 20%. We further found that rainfall may enhance glacier melt contributions to discharge at timescales that complement glacier melt production, possibly explaining why minimum discharge occurred at the study site during warm but dry El Niño conditions, which typically heighten melt in the Andes. Our findings caution against extrapolations from isolated measurements: stream discharge and glacier melt contributions in tropical glacierized systems can change substantially at hourly to interannual timescales, due to climatic variability and surface to subsurface flow processes.

## 1 Introduction

Glaciers supply water resources to over 600 million people worldwide (Messerli et al., 2004). By melting during dry seasons and drought years, they supplement streamflow (Fountain and Tangborn, 1985; Lang, 1986; Escher-Vetter et al., 1994; Jansson et al., 2003; Juen et al., 2007; Soruco et al., 2015; Chen et al., 2017) and ensure reliable water supplies (Mark and Seltzer,

2003; Mark and Mckenzie, 2007; Bury et al., 2013). This has led to the commonly held conceptual model, called the "glacier compensation effect" (Lang, 1986), in which meltwater buffers discharge variability.

Climate change can disrupt the glacier compensation effect, and tropical glacierized watersheds that already experience year-round melt (Kaser and Osmaston, 2002) may be the most vulnerable. Climate models predict amplified temperature increases at high altitudes in low latitudes (Bradley, 2006; Pepin et al., 2015). The retreat of these glaciers temporarily results in increased runoff (Braun et al., 2000; Mark, 2008; Polk et al., 2017; Carey et al., 2017), but gradually depletes the storage of these mountain "water towers". Over time, this reduction in storage capacity can render these glaciers unable to supply sufficient dry-season meltwater discharge for the communities that depend on it (Barnett et al., 2005; Bradley, 2006; Mackay, 2008; Ostheimer et al., 2005; Luce, 2018). Indeed, observations already reveal reduced and fluctuating flows in glacierized watersheds (Mark and Seltzer, 2003; Huss et al., 2008; Baraer et al., 2012; Rabatel et al., 2013; Baraer et al., 2015; Soruco et al., 2015), threatening the water security of millions of people (Immerzeel et al., 2010; Carey et al., 2017; Vuille et al., 2018).

Of all glaciers in the tropics, 99% are located in the Andes (Kaser, 1999), often in remote regions, where resource-limited populations often rely on their meltwater (Bury et al., 2011; La Frenierre and Mark, 2017). Despite over a decade of research in Peru's heavily glacierized Cordillera Blanca (Mark and Seltzer, 2003; Mark and Mckenzie, 2007; Juen et al., 2007; Mark et al., 2005; Baraer et al., 2012, 2015), many of the processes linking variability in climate, glacier melt, and stream discharge remain uncertain. For example, groundwater is also a major contributor to discharge in many glacierized mountainous watersheds around the world (Clow et al., 2003; Liu et al., 2004; Huth et al., 2004; Hood et al., 2006; Tague et al., 2008; Tague and Grant, 2009; Baraer et al., 2009; Andermann et al., 2012; Baraer et al., 2015; Pohl et al., 2015; Somers et al., 2016; Engel et al., 2016; Schmieder et al., 2018; Harrington et al., 2018). This can further modulate discharge through baseflow, but its capacity to do so as glaciers respond to climate change is complicated by largely unconstrained relationships between glacial meltwater and groundwater recharge (Favier et al., 2008; Baraer et al., 2015; Gordon et al., 2015; Minaya, 2016; Harrington et al., 2018).

Understanding how different surface and subsurface pathways influence the timing of meltwater and groundwater contributions to streamflow is critical for predicting how climate change will impact the reliability of watershed discharge. A major challenge in evaluating these spatiotemporal effects in tropical glacierized watersheds is the relative data sparsity and resource limitations in these regions compared to better instrumented mountainous systems in North America and Europe. Many studies in tropical and other remote glacierized settings rely on focused field campaigns using methods such as synoptic water chemistry tracer sampling (Mark and Mckenzie, 2007; Baraer et al., 2009, 2015; Wilson et al., 2016), but these provide only snapshots of the hydrologic state. Even though physically based hydrologic models can provide greater spatiotemporal coverage in mountainous settings (e.g., Suecker et al., 2000; Liu et al., 2004; Tague et al., 2008; Tague and Grant, 2009; Lowry et al., 2010, 2011; Markovich et al., 2016; Pribulick et al., 2016; Omani et al., 2017; He et al., 2018), their application is relatively limited in Andean watersheds (previous implementations include work by Buytaert and Beven, 2011; Minaya, 2016; Omani et al., 2017; Ng et al., 2018) due to the lack of extensive monitoring infrastructure. With these obstacles, there remains limited understanding of how stream discharge in tropical glacierized watersheds varies over time scales ranging from hours to years, and how this variability is driven by dynamic inputs of glacial meltwater and precipitation through a combination of surficial and subsurface pathways.

In this study, we probe the multiple time scales of hydrological processes in a sub-humid glacierized watershed on Volcán Chimborazo in the tropical Ecuadorian Andes. Prior to this work, there have been no comprehensive efforts on Chimborazo to quantify glacier melt as a component of watershed discharge. In contrast to the well-studied crystalline-cored Cordillera Blanca in the outer tropics, Chimborazo is a stratovolcano located in the inner tropics, and therefore experiences less-pronounced

seasonality in precipitation (Kaser and Osmaston, 2002) and more persistent ablation due to higher humidity (Vuille et al., 2003; Favier, 2004; Harpold and Brooks, 2018). Higher humidity can enhance ablation rates by increasing net longwave radiation and condensation (Harpold and Brooks, 2018). Most mixing model analyses of melt contributions in the outer tropics have been limited to the dry season, leaving wet season effects less well understood. In the inner tropics, coincident glacier melt and precipitation inputs throughout the year could lead to multiple processes simultaneously driving discharge variability

that are difficult to disentangle. Furthermore, Andean volcanoes may feature fractured bedrock aquifers that support greater groundwater storage and baseflow than those in crystalline-cored mountainous watersheds (Tague and Grant, 2009; Markovich et al., 2016), adding another factor to be reconciled. A growing body of work at Volcán Antisana, also located in the inner tropics, has begun to shed light on its hydrogeologic (Favier, 2004; Caceres et al., 2006; Favier et al., 2008; Cauvy-Fraunié et al., 2013) and ecohydrologic (Minaya, 2016) conditions, but comprehensive understanding of mountain hydrology in the

inner tropics still greatly lags that in the outer tropics.

Here, we implement field and computational methods to answer two questions: (1) What is the temporal variability of relative glacier melt contributions to discharge, from hourly to multi-year time scales, in a sub-humid glacierized watershed on Volcán Chimborazo? (2) What hydroclimatic factors control this variability? Our approach comprises three methods: mixing model analysis applied to repeat synoptic sampling, time series analysis of hydroclimatic data, and numerical watershed modeling.

Each method interrogates a distinct temporal relationship, and synthesizing their results illuminates how the dominant surface and subsurface processes driving the hydrological response of a tropical glacierized watershed vary as a function of time scale.

## 2   Study Area

Volcán Chimborazo is a glacierized stratovolcano in Ecuador (Figure 1(a)) whose glaciers serve as the headwaters for four major river systems – the Rio Mocha (NE flank), Rio Colorado (NW flank), Rio Guano (SE flank), and Rio Chimborazo (SW

flank) – that supply water to a population of over 200,000 (INEC, 2010). Located in the inner tropics, Chimborazo's climate is characterized by minimal intra-annual temperature variation ($\sim 2°$C) and moderately seasonal precipitation, with two wetter seasons of unequal length (February-May and October-November) (Clapperton, 1990), and two intervening drier seasons that have less but not negligible amounts of precipitation. Moisture mostly originates from the Amazon Basin to the east (Vuille and Keimig, 2004; Smith et al., 2008), which produces a steep northeast (up to 2000 mm/yr) to southwest (<500 mm/yr)

precipitation gradient across the mountain (Clapperton, 1990). Driving interannual climatic variability at the regional scale, El Niño generally brings drier and hotter conditions throughout the Andes (Vuille and Bradley, 2000; Wagnon et al., 2001; Francou, 2003; Bradley et al., 2003; Vuille and Keimig, 2004; Smith et al., 2008), which enhances glacier ablation (Wagnon et al., 2001; Favier, 2004; Veettil et al., 2014b).

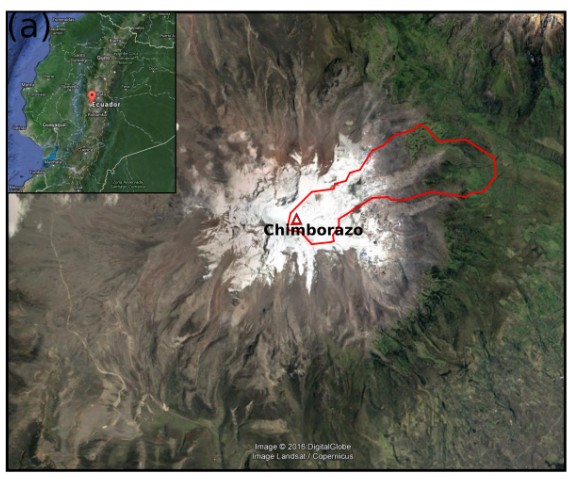

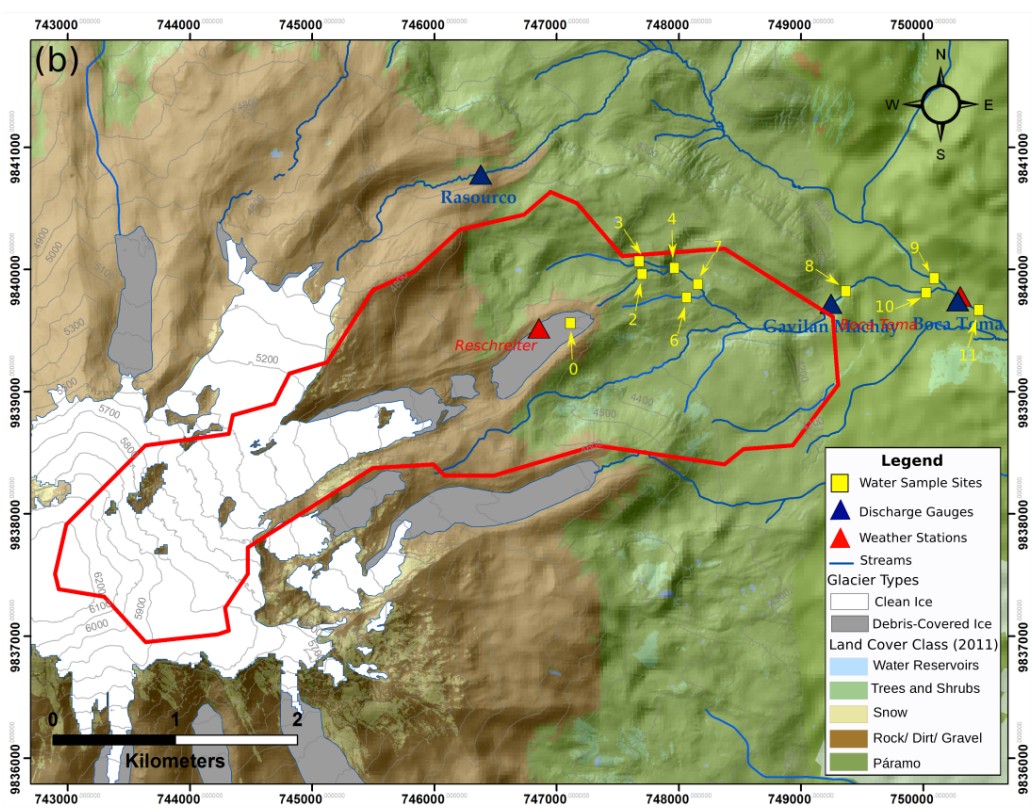

**Figure 1. (a)** Satellite image of Volcán Chimborazo, with the study watershed Gavilan Machay outlined in red, and its location in Ecuador shown in the inset map. The glacierized Gavilan Machay watershed is a relatively humid watershed compared to the western flank of Chimborazo. **(b)** Land cover and locations of monitoring stations and water sampling within the Gavilan Machay watershed.

Records since 1980 indicate that, consistent with the rest of the tropical Andes, temperatures have warmed $0.11°C$ decade$^{-1}$ around Volcán Chimborazo (Vuille et al., 2008; La Frenierre and Mark, 2017). This likely caused the 21% reduction in ice surface area and 180 m increase in mean minimum elevation of clean ice observed between 1986 and 2013 (La Frenierre and Mark, 2017). Although regional precipitation gauges show no notable change over time, local residents report a reduction

in precipitation, which could further drive glacier mass balance changes (La Frenierre and Mark, 2017). Historical records of glacier melt are not available. Under current conditions, only four of Chimborazo's seventeen glaciers, including the two largest, Reschreiter ($2.55$ km$^2$) and Hans Meyer ($1.33$ km$^2$), generate perennial surface discharge, nearly all of which flows northeast into the Río Mocha watershed. The lowest 16% of Reschreiter Glacier is debris-covered, providing insulation that stabilizes ice at lower elevations (4480 m.a.s.l) than would be expected for clean ice, given current climatic conditions. Our

study focuses on the $7.5$ km$^2$ Gavilan Machay sub-catchment on the sub-humid northeast flank of Chimborazo (Figure 1(b)), which is 34% glacierized by Reschreiter and is of concern because it discharges into the main Río Mocha channel just upstream of the Boca Toma diversion point (3895 m.a.s.l. elevation) for an irrigation system.

In addition to glacier melt, groundwater and ecological conditions also control the hydrology of the Gavilan Machay watershed. Springs are prevalent below 4400 m.a.s.l. Geologic maps and stratigraphic interpretations (Barba et al., 2005; Samaniego

et al., 2012) support field evidence for aquifers within unsorted glacial deposits and underlying fractured bedrock (McLaughlin, 2017). Extensive areas of páramo, the biologically rich grasslands endemic to the tropical Andes above $\sim$3500 m.a.s.l., are common across the watershed. Wet páramos commonly contain homogeneous Andosol soils of volcanic origin that can accumulate elevated organic carbon content; this typically gives rise to high porosity, infiltration capacity, hydraulic conductivity, and water retention (Buytaert et al., 2006; Buytaert and Beven, 2011). Absorbent páramo soils are considered to very efficiently

regulate watershed discharge throughout Andean Ecuador (Buytaert et al., 2006; Buytaert and Beven, 2011; Minaya, 2016).

## 3    Methods

### 3.1    Hydroclimatic Data

Precipitation, temperature, and relative humidity data were collected from October 2011 to February 2017 from weather stations installed at 4515 m.a.s.l on the debris-covered portion of Reschreiter Glacier and at 3895 m.a.s.l. at Boca Toma (Figure

S1). Logistical obstacles prevented instrumentation of the upper three-quarters of the watershed above Reschreiter. The Boca Toma weather station was deployed with an Onset Hobo Pendant® Event Logger starting June 16, 2015. The Reschreiter weather station was deployed with an Onset Hobo Micro Station starting October 2011, but temperature and precipitation data recovery was discontinuous. The short data records from Reschreiter were primarily used together with Boca Toma data to determine a lapse rate for precipitation. Precipitation in mountainous watersheds typically exhibit piecewise linear relation-

ships with elevation, with positive (negative) lapse rates below (above) the elevation of maximum precipitation (Wang et al., 2016). We calculated a negative lapse between the two weather stations, which we applied over the entire watershed with the assumption that the elevation of maximum precipitation lies below the watershed. Uncertainty in this approach arises from the actual unconstrained elevation of maximum precipitation (which requires more than two weather stations), and from unquan-

tified precipitation measurement errors that may be caused by wind and freezing temperatures at high elevations. Glacier melt was separately estimated through model calibration to stream discharge data (Section 3.3), which may compensate for errors in the precipitation inputs. Temperature lapse rates were calculated using data collected at glacier ablation stakes (described further below) over June 2016 to November 2016. Relative humidity measured at the Boca Toma station was applied over the entire watershed due to the lack of measurements elsewhere. Discharge simulations should be less sensitive to errors in relative humidity compared to precipitation and temperature, which directly control water inputs to the watershed. We obtained unmeasured meteorological variables (wind speed, solar radiation, longwave radiation, and air pressure) from the Global Land Data Assimilation Systems (GLDAS) (Rodell et al., 2004). A discharge gauging station equipped with a Solinst Levelogger Junior pressure transducer was established at Gavilan Machay, 1.1 km upstream of the Río Mocha confluence. Solinst Barologger measurements at Boca Toma were applied to correct for atmospheric pressure, and standard USGS rating curve techniques (Andrews, 1981a, b) were used to convert water depth to discharge over the period of record (Figure S1).

In June 2016, glacier ablation stakes were installed at two elevations on the Reschreiter Glacier tongue (4792 and 4820 m.a.s.l.) and one on the Hans Meyer Glacier tongue (4925 m.a.s.l.), all in clean ice. Each stake included a temperature sensor, along with a look-down ultrasonic sensor for measuring changes in distance to the ice surface to estimate glacier mass loss in clean ice. The stakes were deployed using the open-source Arduino-based ALog data logger (Wickert, 2014). All sensors were mounted at the top of 3 m long PVC tubes, which were inserted into holes drilled to about 2.5 m depth. In addition to the clean-ice mass loss determined with ablation stakes, glacier volume change of debris-covered ice was estimated by differencing a GPS-validated photogrammetric digital elevation model in 1997 (Jordan et al., 2010) and terrestrial laser scanner (Riegl LMS-Z620) surveys in 2012 and 2013 (La Frenierre, 2016). Because of the sparse spatiotemporal coverage of these glacier melt measurements, these were used only as comparisons for calibrated glacier melt and were not directly applied in our analysis.

Hydroclimatic data collected in the watershed were directly assessed using statistical analyses and implemented as inputs to the integrated hydrologic model. For the statistical analysis, we calculated cross-correlations to probe how the discharge time series may be driven by different climatic factors. Spectral analysis provided further insight on the time scales, represented by time-frequency, over which these interactions occur. Specifically, we examined the magnitude squared coherence ($C_{xy}$) over time-frequency ($f$):

$$C_{xy}(f) = \frac{|S_{xy}(f)|^2}{S_{xx}(f)S_{yy}(f)}, \tag{1}$$

where $S_{xx}(f)$ and $S_{yy}(f)$ are auto-spectral densities of variables $x$ and $y$, respectively, and $S_{xy}(f)$ is the cross spectral density of $x$ and $y$. Like the square of a correlation, the magnitude squared coherence varies between 0 and 1, with 0 indicating the weakest relationship between the two variables at frequency $f$ and 1 indicating the strongest relationship.

### 3.2 Hydrochemical and Isotopic Tracers

#### 3.2.1 Field Sampling

Water samples were collected for use in the Hydrochemical Basin Characterization Model (HBCM) (Section 3.2.3), a hydrochemical mixing model that spans the stream network and requires synoptic water sampling over a sufficiently short time period such that data reflect spatial and not temporal variability. We carried out five synoptic sampling campaigns during January 1-8, 2012; July 7-9, 2012; June 12-15, 2015; June 25-30, 2016; and February 4-7, 2017. The June and July (January and February) samples represent the longer (shorter) dry season. Dry seasons were targeted because of water resource interests during these periods; integrated hydrologic model simulations served to extend the analysis to dry seasons. In addition to limiting the number of days spanned during a sampling campaign, synoptic sampling should avoid hourly timescale hydrochemical fluctuations. All samples were collected between mid-morning and mid-afternoon. In February 2017, we confirmed that 1-minute resolution specific conductivity changes over a 24-hour time period at the Reschreiter glacier tongue were an order of magnitude smaller than the spatial variability across the Gavilan Machay subcatchment (details in McLaughlin, 2017). Logistical difficulties prevented similar measurements farther downstream in the watershed, where dynamic melt versus groundwater contributions likely caused greater hydrochemical variability (Section 4.1).

During each of the five campaigns, we collected water samples from meltwater, which may contain both glacier melt and snowmelt, springs, and precipitation, as well as at stream confluence mixing points (locations shown in Figure 1(b)). Spring samples from concrete capture boxes or natural valley wall seeps represent groundwater, which consists of an unconstrained mix of shallow saturated soil water from páramo areas, morainic debris aquifer water, and deeper fractured bedrock aquifer water. Precipitation samples were collected using evaporation-proof totalizing rain gauges deployed for 3-6 days at Boca Toma, near the Reschreiter weather station, and near Hans Meyer glacier (at 4780 m.a.s.l.). Each field campaign covered most of the same sampling locations between the Reschreiter glacier tongue and the Gavilan Machay confluence. The 2012 and 2017 sampling periods included additional stream samples between some confluences to estimate groundwater contributions along shorter stream reaches (Figure 2(b)). For each sampling site, 30mL of water were collected, filtered in the field using either 0.45 $\mu$m (before 2017) or 0.2 $\mu$m (2017) filters, and stored in Nalgene bottles that were capped and sealed with electrical tape, and then stored near 4°C as soon as possible.

#### 3.2.2 Laboratory Analysis

In 2012, major dissolved ions ($Li^+$, $Na^+$, $K^+$, $Mg^{2+}$, $Ca^{2+}$, $F^-$, $Cl^-$, $NO_3^-$, $PO_4^{3-}$ and $SO_4^{2-}$) were measured using a Dionex DX500 Ion Chromatographer at the Water Isotope and Nutrient Laboratory at The Ohio State University, and stable isotopes of water ($\delta$18O and $\delta$2H) were measured using Piccaro L2130-i CRDS isotope analyzers at the Water Isotope and Nutrient Laboratory and at the Byrd Polar Research Center. In 2015-2017, cations ($Na^+$, $K^+$, $Mg^{2+}$, $Ca^{2+}$) were measured using an Agilent 7700X Inductively Coupled Plasma-Mass Spectrometer (ICP-MS) at Gustavus Adolphus College, anions ($F^-$, $Cl^-$, $SO_4^{2-}$) were measured using a Dionex ICS1000 Ion Chromatographer (IC) also at Gustavus Adolphus College, and stable isotopes of water ($\delta^{18}O$ and $\delta^2H$) were measured at the University of Minnesota using an LGR DLT-100 Liquid Water Analyzer (a laser spec-

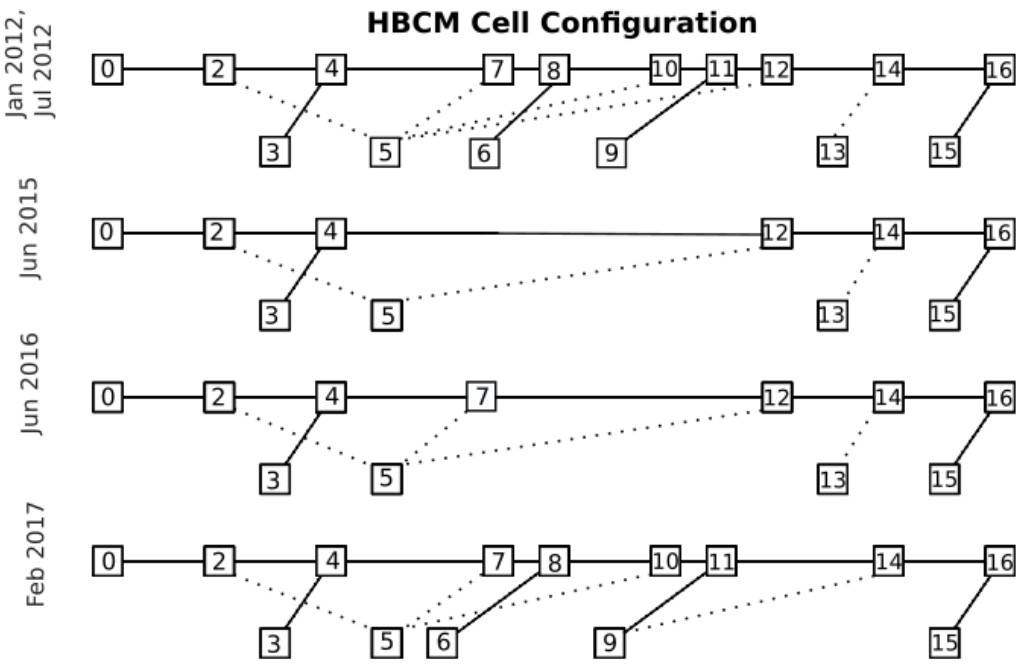

**Figure 2.** Different computational cell configurations for the HBCM mixing model based on the available samples (site codes in squares, see Figure 1) for each of the five periods. The upper solid line for each period represents the main channel. The lower solid lines depict tributary links for confluence cells, and the lower dotted lines show groundwater inputs to the main channel for reach cells.

troscopy system). We calculated the bicarbonate ($HCO_3^-$) concentration as the charge balance residual. Reported isotope ratios are relative to the Vienna Standard Mean Ocean Water (VSMOW) and typical precisions are $\pm 1.0‰$ for deuterium/hydrogen values and $\pm 0.25‰$ for $^{18}O/^{16}O$ values. Checking for consistency across instruments, we confirmed that bulk concentrations at each location and spatial trends for each analyte were similar across sampling periods (Figure S3). Certain analytes did exhibit
5   a discernible systematic bias for one particular sampling period compared to other periods that may be related to laboratory instrument, but this should not pose a problem for the mixing model, which is implemented only using data within the same sampling period (measured on the same instrument).

### 3.2.3    Mixing Model: Hydrochemical Basin Characterization Model (HBCM)

Naturally occurring dissolved ions and stable isotopes of water ($\delta^{18}O$ and $\delta^2H$) have long been used to track the relative
10   contributions of different surface source waters to total watershed discharge (Hooper and Shoemaker, 1986; Mark and Seltzer, 2003; Ryu et al., 2007; Mark and Mckenzie, 2007; Baraer et al., 2009), as well as to identify groundwater flow paths (Clow et al., 2003; Kendall et al., 2003; Baraer et al., 2009; Crossman et al., 2011; Baraer et al., 2015). Here, the proportion of glacier and snow melt versus groundwater in discharge at the Gavilan Machay watershed is quantified using the Hydrochemical Basin Characterization Model (HBCM), a multi-component hydrochemical mixing model developed for use in data-sparse,

glacierized tropical watersheds (Baraer et al., 2009). Given source (or "end-member") and outflow chemistries at different mixing points throughout the watershed, HBCM solves an over-constrained set of mass balance equations for multiple tracers to determine the relative flow contributions of each source. Details are provided in the Supplementary Information (HBCM Section).

As with all hydrochemical mixing models, HBCM's calculation of relative source contributions depends on three fundamental assumptions: (1) end-member chemistry is unique and spatially homogeneous within the analysis area, (2) tracers are chemically conservative within the analysis, and (3) end-member mixing is instantaneous and complete (Christophersen et al., 1990; Soulsby et al., 2003). A unique feature of HBCM is that it represents spatial information within a watershed through a series of cells that are interconnected by having outflow from one become inflow to a subsequent downstream cell. There are

two types of cells, both of which have streamflow at the upgradient end of the cell as a source and streamflow at the downgradient end of the cell as the mixed water output. "Reach cells" have groundwater as their other source, and "confluence cells" have tributary water as their other source (see cartoon in Figure S2). Note that this makes end-members for a particular HBCM cell different than for the full watershed, which has only meltwater and groundwater as sources contributing to discharge at the outlet. Figure 2b shows the conceptual schematics of the Gavilan Machay cell configuration for the five sampling periods.

Although HBCM only requires the three assumptions to be met on a cell-scale, we carried out a preliminary watershed-level analysis considering groundwater and meltwater as sources to all mixed stream samples, in order to identify potential conservative tracers that are reasonable candidates for all cells. Appropriate tracers should show end-members appearing on opposite ends of a line formed by mixed samples in bivariate plots, and samples for different end-members should group separately from each other in hierarchical cluster analysis diagrams (Christophersen et al., 1990; Hooper, 2003; James and

Roulet, 2006). Stable isotopes were excluded as tracers for reach cells, because the groundwater likely has a range of isotopic values due to different recharge elevations.

    To bracket some of the uncertainty in the method, HBCM generates estimates of fractional contributions from each source using different combinations of potential tracers. The final result consists of a range of estimates that produce similar (within about three times the minimum) cumulative residual errors between the measured tracer concentrations in the mixed outflow

water and that predicted by the over-constrained mixing model. This quantifies uncertainty due to the model's inability to distinguish among equally good optimization results but represents only a lower limit of error, because it does not account for the mismatch between the observed and predicted mixed concentration outflows. There are no straightforward methods to convert the sum of residual concentration flux errors to estimated source contribution errors.

### 3.3   Integrated Hydrologic Modeling

Spatially distributed watershed models can integrate surface hydrology and groundwater flow through time to evaluate their joint impacts on water resources. Over the one-year period of June 2015-June 2016 when continuous air temperature and precipitation measurements are available in the watershed, we implemented Flux-PIHM (Shi et al., 2013), an intermediate complexity watershed model that balances mechanistic parameterizations with computational efficiency. Full details about Flux-PIHM can be found in Qu and Duffy (2007) and Shi et al. (2013); here, we summarize the major features. Flux-PIHM

couples physically based equations for canopy interception, infiltration, surface and subsurface water flow, and snow melt with the energy balance scheme of the NOAH land-surface model (Ek et al., 2003) for more accurate simulation of evapotranspiration. Flux-PIHM employs a semi-discrete finite volume approach on an unstructured grid that performs efficiently on steep topographies. Channel and overland flow are represented by diffusion wave approximations to St. Venant equations; shallow

groundwater flow follows a 2D Dupuit approximation; and unsaturated zone flow is based on a 1D form of Richards equation. The model simulates water storage in one vertically integrated unsaturated zone layer and one vertically integrated saturated zone layer, providing a "2.5D" distributed model.

Flux-PIHM determines snowmelt based on energy balance. Use of coarse-scale GLDAS radiation inputs introduces errors, but as will be discussed in Section 4.3.1, precipitation limitations may make snowmelt calculations less sensitive to radiation

input uncertainties compared to glacier melt calculations. Due to the unavailability of high resolution radiation input measurements as well as intensive source-code modifications required to couple energy balance calculations for ice melt into the Flux-PIHM, we added a separate module to simulate glacier melt using a temperature index scheme (NRCS, 2009). Although the accuracy of a temperature index glacier melt model for tropical glaciers can be uncertain due to uncaptured effects of solar radiation, cloud cover, humidity, topography, and aspect (Hock, 1999, 2005; Pellicciotti et al., 2005; Sicart et al., 2008; Huss

et al., 2009; Gabbi et al., 2014; Fernández and Mark, 2016), it remains the most feasible approach in poorly instrumented watersheds given its simplicity and limited field data requirement compared to an energy balance approach (Hock, 2005; Fernández and Mark, 2016; Reveillet et al., 2017). The temperature index glacier melt model includes:

$$F_I = M_I(T_a - T_{M,I}) \qquad T_a > T_{M,I} \tag{2a}$$

$$F_I = 0 \qquad T_a < T_{M,I} \tag{2b}$$

where $F_I$ is the ice melt rate (m/hr), $T_a$ is air temperature in the grid cell containing ice (°C), $M_I$ is the melt factor parameter (m/hr/°C) to be calibrated, and $T_{M,I}$ (°C) is set to 0°C as the air temperature threshold for ice to melt. Over the simulated time period, we assume that there is an unexhausted supply of ice that can melt in the glaciated grid cells below the equilibrium line

altitude (ELA) at 5050 m.a.s.l. (Frenierre and Mark, 2014), which is a reasonable approximation over the one-year simulation period. The melt simulated with the temperature index model was added to the precipitation amount for the Flux-PIHM forcing inputs.

We used the PIHMgis software (Bhatt et al., 2014) to construct an unstructured domain of 188 cells over the Gavilan Machay subcatchment using a 30 m resolution Shuttle Radar Topography Mission (SRTM) Digital Elevation Model (DEM) (Farr et al.,

2007). Although a major feature of PIHMgis is its tight integration with spatial and temporal datasets for model inputs such as soil properties and meteorological forcing, these datasets only cover densely monitored regions, mostly within North America and Europe. For meteorological forcing, we used the spatially distributed inputs described in Section 3.1. Vegetation mapping by McLaughlin (2017), based on 30-cm resolution aerial photo surveys conducted by the Sistema Nacional de Información de Tierras Rurales e Infrastructura Tecnológica (SIGTIERRAS; http://www.sigtierras.gob.ec/descargas/), provided land-cover

types and boundaries. Built-in land-cover parameters from Noah-LSM were used for the "grassland/herbaceous" type at lowest elevations corresponding to páramo, the "barren/sparsely vegetated" type for intermediate elevations with rock/dirt/gravel, and the "perennial ice/snow" type for the ice-covered areas. This approach simplifies the mix of tussock grasses, acaulescent rosettes, and cushion plants that make up the páramo into a single representative "grassland/herbaceous" type in order to

reduce the calibration burden. For the grassland/herbaceous land cover type, the default monthly Leaf Area Index (LAI) values were replaced with measurements from MODIS (Vermote, 2015) to avoid using incorrect seasonal changes from the original model settings for this tropical region. Hydraulic parameters were manually calibrated to match observed discharge at Gavilan Machay.

## 4    Results and Discussion

This section presents the respective insights gained from each of the three methods on the temporal relationship among meltwater, groundwater, and discharge: the mixing model analysis offers discrete multi-year estimates over five years, the time series analysis shows fine-scale hourly resolution correlations, and the integrated hydrological model explores intermediary weekly to seasonal processes within a one-year simulation period containing a strong El Niño event. A complete interpretation of the multi-scale temporal variabilities and their hydroclimatic controls emerges in Section 4.3.2 when evaluating the model

simulations in relation to the mixing model and time series analysis results.

### 4.1    Mixing-model analysis of meltwater contributions to discharge

Total cation concentrations provide a summary representation of hydrochemistry results from the five dry-season synoptic sampling periods in Figure 3. These plots show that even though hydrochemical conditions vary over the different periods, groundwater samples, which geochemically interact with soil and rocks, consistently contain much higher ion concentrations

than meltwater samples. The distinctive chemistries of groundwater and meltwater make it possible to use the mixing model approach to estimate their relative contributions to streamwater, which shows an increase in ion concentration while moving downgradient due to the cumulative addition of groundwater (see Figure 1(b) for sample locations). We chose as tracers those analytes that most consistently showed the mixed sample visually falling close to the line between its two source samples in the bivariate plots in Figures S4-S8: sum of monovalent cations, $Mg^{2+}$, $Ca^{2+}$, $Cl^-$, and $HCO_3^-$. Hierarchical cluster analysis lends

confidence that these five potential tracers can be used in the mixing model analysis to distinguish between groundwater and melt samples as different watershed-level end-members (Figure S9). Precipitation was not included as an end-member, because precipitation samples tended to plot outside the range of stream samples bracketed tightly by groundwater and meltwater samples in bivariate plots. Rather than directly add to streamflow, any precipitation that fell close to the sampling time likely evapotranspired or infiltrated and contributed to streamflow through groundwater.

HBCM results in Figure 4 illustrate the importance of both meltwater and groundwater in the watershed. Surficial meltwater comprises between 23-66% of discharge during the five dry-season sampling periods, with groundwater constituting the remaining 34-77% at any given time. Notable differences were observed across the sampling periods. The higher relative melt

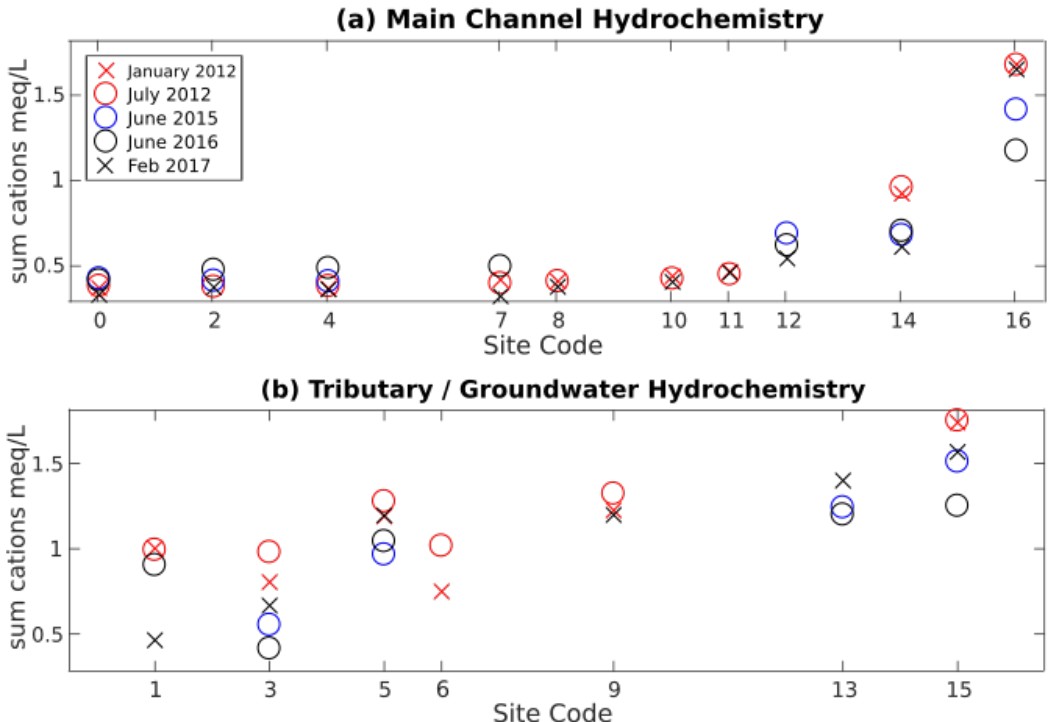

**Figure 3. (a)**Hydrochemistry at different sampling locations along the Gavilan Machay main channel shows variability in space and over the five sampling periods; site codes are ordered from highest (0 for meltwater) to lowest (16 for Boca Toma) elevation (see Figure 1(b)). **(b)** Hydrochemistry for different tributary (sites 3, 6, 9, and 15) and groundwater (spring) samples (sites 5 and 13), ordered from highest to lowest elevation, show variability in space and time.

contribution during February 2017 compared to January 2012 could reflect the accelerating melt rates observed on Chimborazo (La Frenierre and Mark, 2017). However, the absolute melt contribution, determined by applying estimates of relative melt contributions to average observed weekly discharge measurements around the sampling time, was in fact lowest in February 2017, because of significantly less total discharge compared to the other sampling periods (Figure 4(b)). The lower total discharge

5   was likely due to lower precipitation and temperature during the weeks around the sampling period compared to during the other sampling periods (Figure S1). Our findings across the five sampling periods demonstrate that one single synoptic tracer test should not be directly generalized or interpreted without considering temporal dynamics and groundwater conditions.

Our results show Volcán Chimborazo to deviate from trends found at the well-studied Cordillera Blanca, likely due to its distinct climatic and geologic conditions. When compared to an exponential fit between relative groundwater contribution

10   and glacierized fraction for four watersheds in the Cordillera Blanca (Baraer et al., 2015), our estimates for groundwater contributions in Gavilan Machay are approximately twice as large. Also, the glacierized Gavilan Machay sub-catchment of the Upper Río Mocha watershed has a specific discharge that is less than half of that in the non-glaciated portion, in contrast to

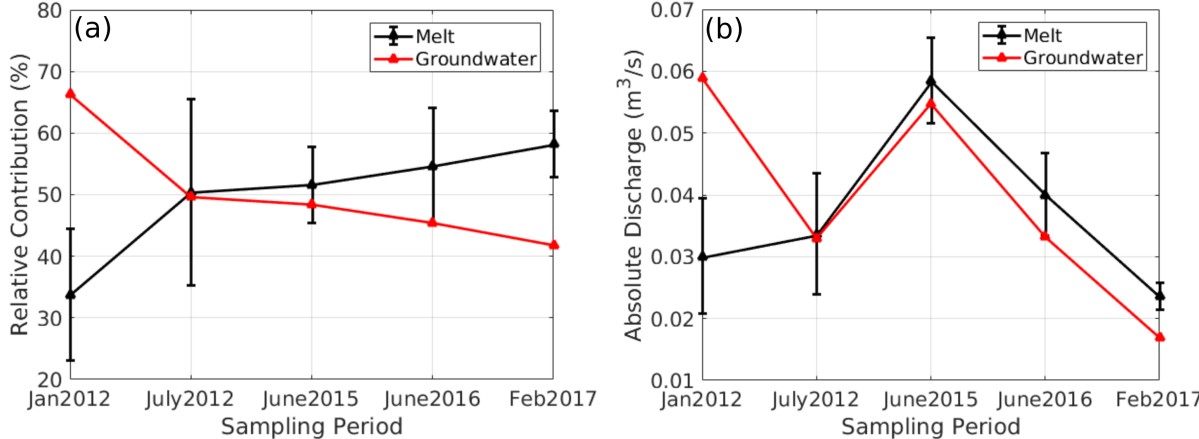

**Figure 4.** The HBCM mixing model predicts a range in relative surficial meltwater contributions to discharge over five discrete sampling times, which may reflect both temporal changes and uncertainties. Error bars bracket HBCM estimates that produced similar best matches to observed tracer concentrations; however, actual uncertainties are higher because of residual errors. Absolute meltwater discharge contributions can vary in time very differently than relative inputs, in part due to varying groundwater contributions.

the greater specific discharge generally found with greater glacierized areas in the Cordillera Blanca (Baraer et al., 2009; Mark and Seltzer, 2003).

HBCM implementation with five different field campaigns enabled us to evaluate uncertainties due to distinct sampling plans (details in Supplementary Information - Table S1). We found that changing HBCM cell configurations could generate up to

5  a 23% melt fraction difference in estimates. Also, having fewer HBCM analysis cells (e.g., longer stream reaches) and fewer groundwater samples consistently led to greater HBCM residual errors. Too few groundwater samples becomes problematic when groundwater is not a homogeneous end-member throughout the watershed, which is the case in Gavilan Machay, which contains springs with somewhat higher solute concentrations at lower elevations (Figure 3(b)). Further, errors in the estimated groundwater contribution grow when using fewer and longer reach cells, because with additional and shorter reach cells,

10  observations can reset the stream channel chemistry to correct concentrations. These results demonstrate the importance of adequately measuring the spatial variability of the surface and subsurface flow network, and they prompt the use of alternative methods to help constrain uncertainties in HBCM analysis results.

### 4.2 Time series and spectral analysis of hydroclimatic controls

Because of the uncertainties and long time gaps in the HBCM analyses, we applied statistical analyses to the continuous data

15  available from July 2015 to March 2016 at the Boca Toma weather station and Gavilan Machay gauging station to further infer characteristic trends between meltwater and discharge and their climatic controls. Considering air temperature as a proxy indicator of meltwater, the hourly cross correlation of air temperature leading discharge at Gavilan Machay in Figure 5(a)

shows a strong diurnal signal, with peak discharge occurring four hours after the warmest part of the day at an average rate (0.1 m³/s) that is about twice the magnitude of average morning discharge.

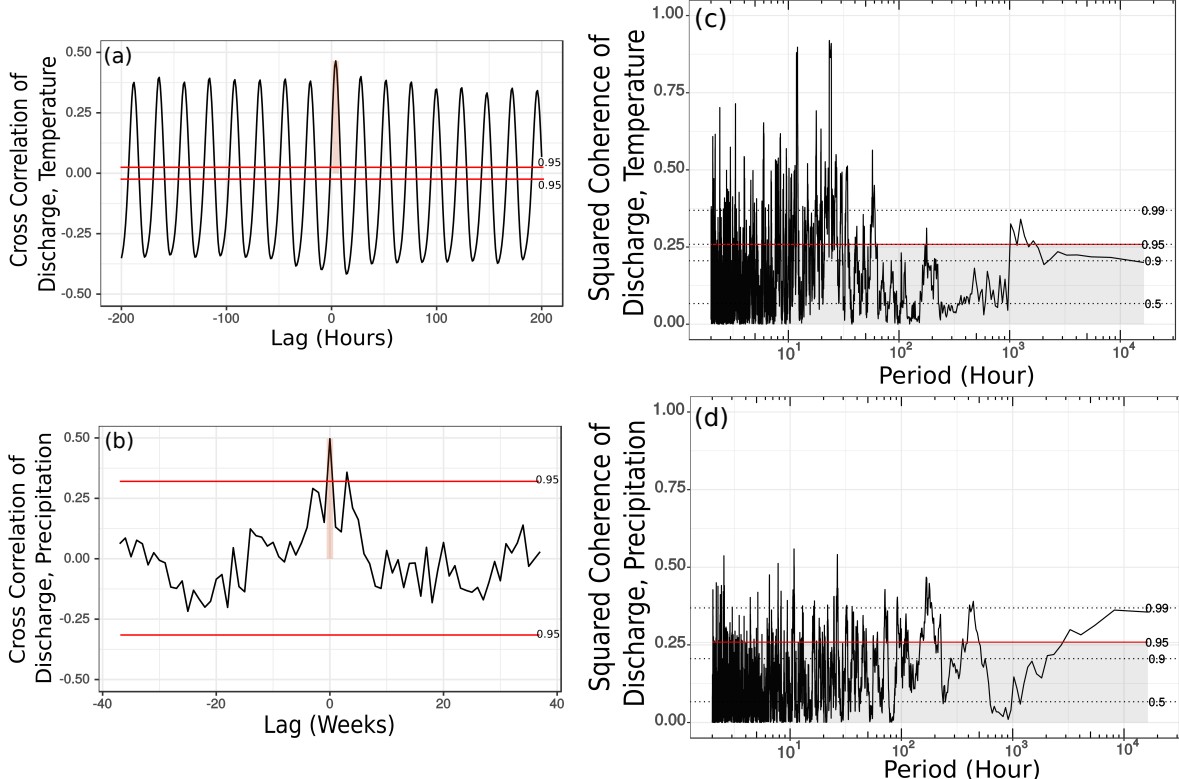

**Figure 5.** Cross-correlations (with 95% confidence interval shown) of observed discharge at Gavilan Machay with **(a)** hourly temperature and **(b)** weekly precipitation show that discharge has a clear diurnal link with temperature at about a 4 hour lag and a strong relationship with weekly precipitation, respectively. Magnitude squared coherence (with various confidence intervals shown) between discharge and **(c)** temperature exhibits a high peak at 24 hours corresponding to the cross-correlation result, as well as a strong peak at 12 hours, and a more moderate peak at a multi-day scale. The coherence between discharge and **(d)** precipitation peaks between 100 to 200 hours (about 1 week) and may also be significant at scales approaching one year.

To determine if melt could be driving discharge variability beyond diurnal time scales, we examined the magnitude squared coherence between temperature and discharge in Figure 5(c), which quantifies their correlation at a certain period (inverse time-frequency). As expected from the time-series results, the most prominent feature is the peak at a period of 24 hours. Interestingly, the next highest coherence is at a period of 12 hours. While this could be a harmonic artifact of the dominant 24-hour trend, it is supported by a slight temperature increase commonly observed around 11pm. The resulting increase in discharge, while detectable, is inconsequential to the total daily water balance. Other very narrow coherence peaks at periods less than 12 hours are likely spurious, because the power spectral densities of both temperature and discharge are low over that

range (Figure S10). However, the smaller but broader peak around 50 to 60 hours suggests that multi-day warming may also drive multi-day discharge events, though this link is much weaker than the diurnal response.

The substantial groundwater contributions to discharge inferred from the HBCM analysis prompts a look at not only melt but also precipitation controls on multi-day discharge. Hourly precipitation and discharge are very weakly correlated (Figure S11(a)); however, a significant correlation of 0.5 appears for weekly averages with zero time lag (Figure 5(b)). Correspondingly, a high (above 95% confidence interval) and broad coherence peak between precipitation and discharge can be seen over periods of about one week (168 hours) (Figure 5(d)). Together, these results suggest that sustained rain events influence discharge over a week, and therefore that rainwater tends to infiltrate instead of flow quickly overland. Other statistically significant (at the 95% confidence interval) coherences between discharge and both temperature and precipitation across multi-week to multi-month periods further support the role of even slower subsurface flow pathways.

Combining the time series analysis with the HBCM results suggests that streamflow at Gavilan Machay is heavily influenced by both surficial meltwater and groundwater, and that the latter is driven by precipitation. Furthermore, time series and spectral analyses highlight temporal links not easily found through a fieldwork-intensive tracer-based approach: meltwater feeds discharge at Gavilan Machay on an hourly time scale, while weekly discharge responds most strongly to precipitation events. There are, however, limitations to this statistical assessment that result from the short nine-month dataset, which precludes robust examination of any seasonal to multi-year responses to bimodal wet and dry seasons and El Niño effects. Hydrologic modeling in the following section can address this, as well as questions about the quantitative role of melt contributions or groundwater buffering periods of low rainfall.

## 4.3 Integrated Hydrologic Model Simulations

### 4.3.1 Calibration Results

Matching the observed discharge dynamics with the hydrologic model required calibration of two different melt factors based on time period: a lower value of 7.10 mm.w.e. (millimeters water equivalent) $C^{-1}$ day$^{-1}$ over December 2015-February 2016 and a higher value of 8.64 mm.we.$°C^{-1}$ day$^{-1}$ over the rest of the simulation. The estimated melt factors fall within the range of melt factors calculated for other glaciers in the tropics (3.5-9.9 mm.we.$°C^{-1}$ day$^{-1}$) reported in Fernández and Mark (2016). Our simulation period coincides with a strong El Niño event that generated the warmest and driest conditions from late November 2015 to the start of February 2016 (Figure 6(a)–(b)). The lower melt factor at this time dampens the intensity of glacier melt, possibly because of the absence of heat transfer from rain (Francou, 2004), but it nonetheless simulates among the highest glacier melt volumes of the simulation period (Figure 6(a)), consistent with other studies showing increased glacier melt in response to El Niño events in the Andes (Francou, 2004; Veettil et al., 2014a; Manciati et al., 2014; Maussion et al., 2015; Veettil et al., 2016). Huss et al. (2009) similarly estimated a lower melt factor in the Swiss Alps under warmer conditions, which they attributed to warmer conditions being driven by longwave rather than shortwave radiation. Overall, in Gavilan Machay, the average specific glacier melt rate (in water equivalence) simulated over glaciated areas below the ELA was 1.5 m/yr. This falls within the range measured at the Reschreiter Glacier tongue, bracketed by mass balance estimates on slower-melting debris-

covered ice of 0.87 m/yr (1997–2013) and 0.54 m/yr (June 2012–January 2013), and average ablation stake observations on faster-melting clean ice of 3.4 m/yr (June–November 2016). Although useful for comparing against the calibrated melt model, these measurements do not cover sufficient areas and periods of time to constrain separate melt factors for debris-covered and clean-ice melt factors.

Over the entire watershed, the resulting calibrated glacier melt production is equivalent to 68% of the precipitation input and 567% of the simulated snowmelt amount during the simulation period. Based on temperature, Flux-PIHM partitioned the precipitation input into 12% snowmelt and 88% rainfall. The much smaller amount of simulated snowmelt compared to glacier melt supports the earlier suggestion (Section 3.3) that snowmelt could be precipitation-limited rather than energy-limited. This helps justify our separate approach of simulating snowmelt through Flux-PIHM's energy balance module while simulating

glacier melt through a calibrated temperature-index model, because energy balance calculations of glacier melt would be much more sensitive to uncertainties in coarse-scale radiation inputs than snowmelt. We do acknowledge, however, that snowmelt simulations depend on lapse-rate-determined precipitation inputs that have their own uncertainties (Section 3), and so our calibrated glacier melt may include some amount of snowmelt that is not represented in the model.

The calibration procedure also involved soil parameter adjustments. Hydraulic parameter estimates in páramo environments

are scarce, and their characterization can be uncertain (Buytaert et al., 2006). For an initial estimate, we applied pedotransfer functions used in Flux-PIHM (Wosten et al., 1999) to a range of páramo soil measurements from a study area 20 km northwest of Chimborazo (3800 to 4200 m.a.s.l.) (Podwojewski et al., 2002) and from a study watershed on glaciated Volcán Antisana also in the Ecuadorian Andes (4000-4600 m.a.s.l.) (Minaya, 2016) (see Table S2 for full details). We then calibrated the model for three mapped land-cover zones corresponding to páramo, rock/dirt/gravel, and ice (Figure 1). In the páramo zone (Table 1),

matching observed discharge required lower hydraulic conductivity and greater water retention (expressed in van Genuchten hydraulic parameters) than initially estimated. This is likely due to high organic matter content supported by the study area's sub-humid conditions and the well-recognized retentive hydraulic properties of páramo soils (Podwojewski et al., 2002; Buytaert et al., 2006). In the sparsely vegetated and ice-covered zones, the calibration yielded higher hydraulic conductivities and lower water retentions than in the páramo zone, corresponding to reduced organic matter fraction and fractured bedrock, though

the hydraulic conductivities were still lower than the initial estimates from the pedotransfer functions.

Simulation results in Figure 6(c) show that the calibrated model parameters closely produced the observed weekly discharge, including lower discharge under the drier and warmer El Niño conditions in December 2015-January 2016. The single major model mismatch occurred at a precipitation-driven discharge peak in February 2016. This could reflect uncertainties from the soil parameter calibration, as well as from our use of a lapse rate-based precipitation field over complex terrain, in which

high-altitude precipitation events may not all be recorded at the low-altitude rain gauge. On shorter timescales, hour-of-day simulation results in Figure 7(b) demonstrate that the model does produce a diurnal trend, but with slightly less than half the average range and at a 6-hour later peak compared to observations (Figure 7(a)). These hourly discrepancies can be attributed to weaknesses in the simple melt model. Hock (2005) argued that temperature index models can successfully capture seasonal glacier melt trends but struggle with diurnal fluctuations, which are strongly driven by solar radiation dynamics. Although our

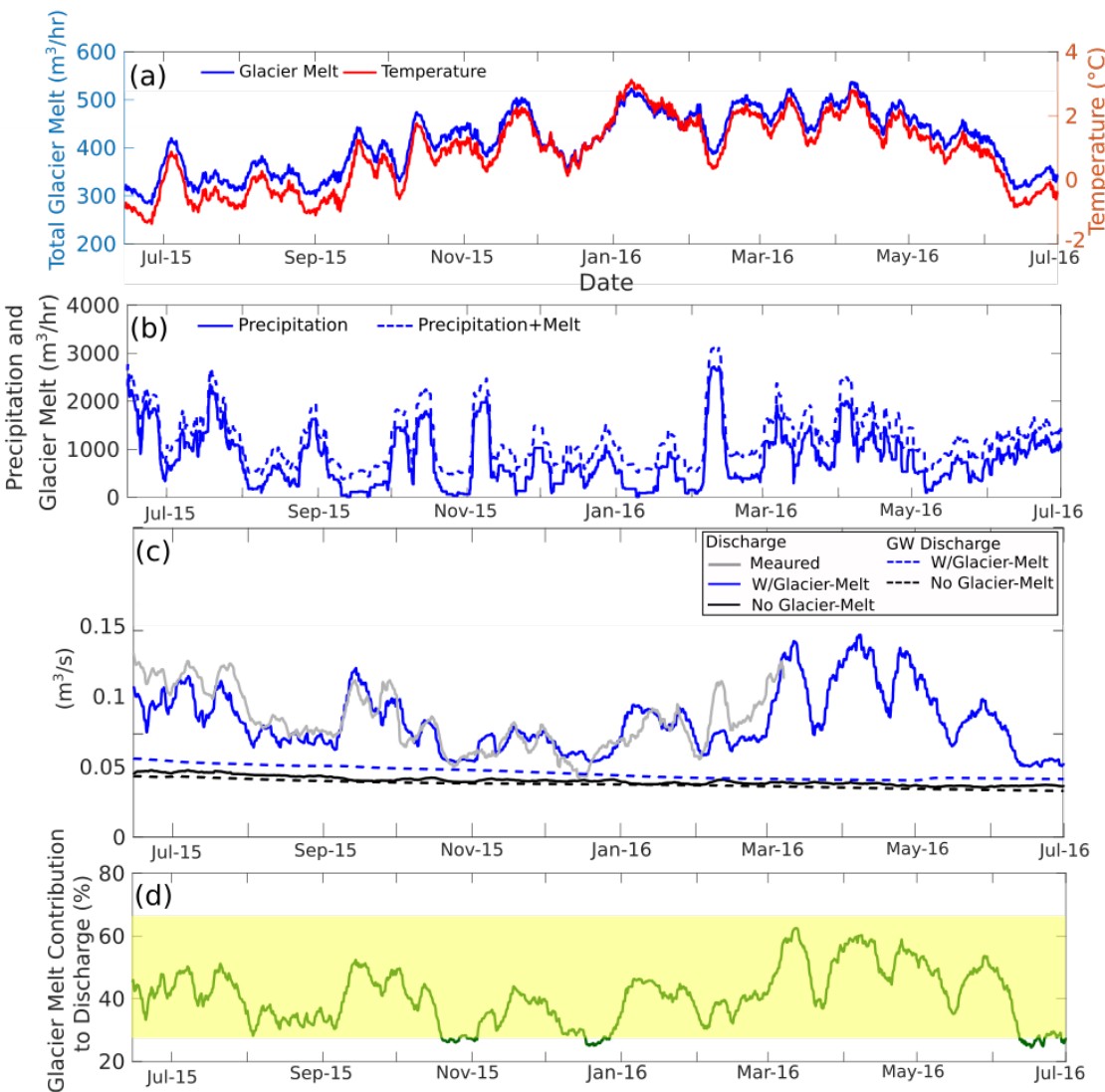

**Figure 6.** Time series of weekly moving average of **(a)** average air temperature over the ablation zone (glacier-covered areas below the ELA (5050 m.a.s.l.)) and simulated glacier melt production; **(b)** precipitation (solid line), and precipitation+glacier melt (dashed lines) production **(c)** discharge at Gavilan Machay from observations (gray), calibrated simulations (blue solid), and simulations with no ice-melt (black solid); groundwater contribution to discharge for calibrated (blue dashed) and for no-ice simulations (black dashed); and **(d)** simulated percent glacier melt contribution to discharge at Gavilan Machay. The shaded block shows the range of percent melt contributions estimated for five sampling periods over 2012-2017 from the mixing model; these estimates may include snowmelt in addition to glacier melt, but omit meltwater contributions through groundwater.

simulations cannot reliably produce the timing of hourly discharge, they can provide informative lower bounds on the size of the average diurnal range.

| | KINF (m/s) | KSATV (m/s) | KSATH (m/s) | Porosity | Residual Moisture | $\alpha$ (1/m) | $\beta$ (-) |
|---|---|---|---|---|---|---|---|
| Range predicted for observed páramo soil textures* | 3.71E-8–8.28E-5 | 8.42E-8–3.71E-5 | 8.42E-7–6.96E-5 | 0.418–0.493 | 0.05 | 0.327–5.82 | 1.1–1.173 |
| Calibrated: Grassland$^\diamond$ | 1.23E-07 | 4.02E-08 | 4.02E-07 | 0.458 | 0.05 | 0.488 | 1.066 |
| Calibrated: Sparsely vegetated$^\diamond$ | 1.43E-07 | 4.63E-08 | 4.63E-07 | 0.459 | 0.05 | 0.585 | 1.063 |
| Calibrated: Ice-covered$^\diamond$ | 2.07E-07 | 6.71E-08 | 6.71E-07 | 0.461 | 0.05 | 0.863 | 1.06 |

*Predicted using pedotransfer functions with 9 páramo soil texture measurements from Ecuador. Samples are from 3 locations in a glaciated watershed in Volcan Antisana (Minaya, 2016) and 4 locations about 20 km northwest of Chimborazo (Podwojewski et al., 2002) (see Supplementary Information for more details).

$^\diamond$Calibrated to match observed discharge at Gavilan Machay

**Table 1.** Soil hydraulic parameters calibrated for the three soil type areas compared against the range predicted using the pedotransfer function with measured páramo soil textures in Ecuador (Podwojewski et al., 2002; Minaya, 2016). Parameters include hydraulic conductivities for vertical infiltration (KINFV), vertical saturated zone flow (KSATV), and horizontal saturated zone flow (KSATH); porosity; residual soil moisture; and shape parameters ($\alpha$ and $\beta$) for the van Genuchten moisture retention curve: $\theta = \theta_{res} + porosity \times \left(\frac{1}{1+|\alpha\psi|^\beta}\right)^{(1-\frac{1}{\beta})}$, with water content $\theta$ and pressure head $\psi$.

### 4.3.2 Simulations of relative glacier melt contribution to discharge

To quantify relative glacial meltwater contribution to stream discharge using Flux-PIHM, we compared the calibrated discharge simulations at Gavilan Machay ($Q_{Calib}$) with simulations that omitted glacier meltwater in the forcing inputs ($Q_{NoGlacierMelt}$). We then calculated relative glacial meltwater contribution to discharge via:

$$\%GlacierMelt = \frac{Q_{Calib} - Q_{NoGlacierMelt}}{Q_{Calib}} \times 100\%. \tag{3}$$

Apart from the addition of glacier melt in the calibrated case, the two model scenarios include the same model inputs, including air temperature, precipitation (including identical snowmelt), land cover, and hydrologic properties. Thus, our calculation of change between the two scenarios isolates the effect of having glacier melt versus not having glacier melt. Overall, over the one-year simulation period, an average 52% of stream discharge in Gavilan Machay can be attributed to glacier melt, compared to an 8% contribution by snowmelt. Because the simulated glacier melt amount was calibrated in addition to precipitation inputs, we consider this glacier melt contribution to originate from the pre-existing ice reservoir at the start of the model period. However, as noted above (Section 4.3.1), the calibrated glacier melt contribution could include some amount of snowmelt not fully accounted for in the model due to uncertainties in precipitation inputs.

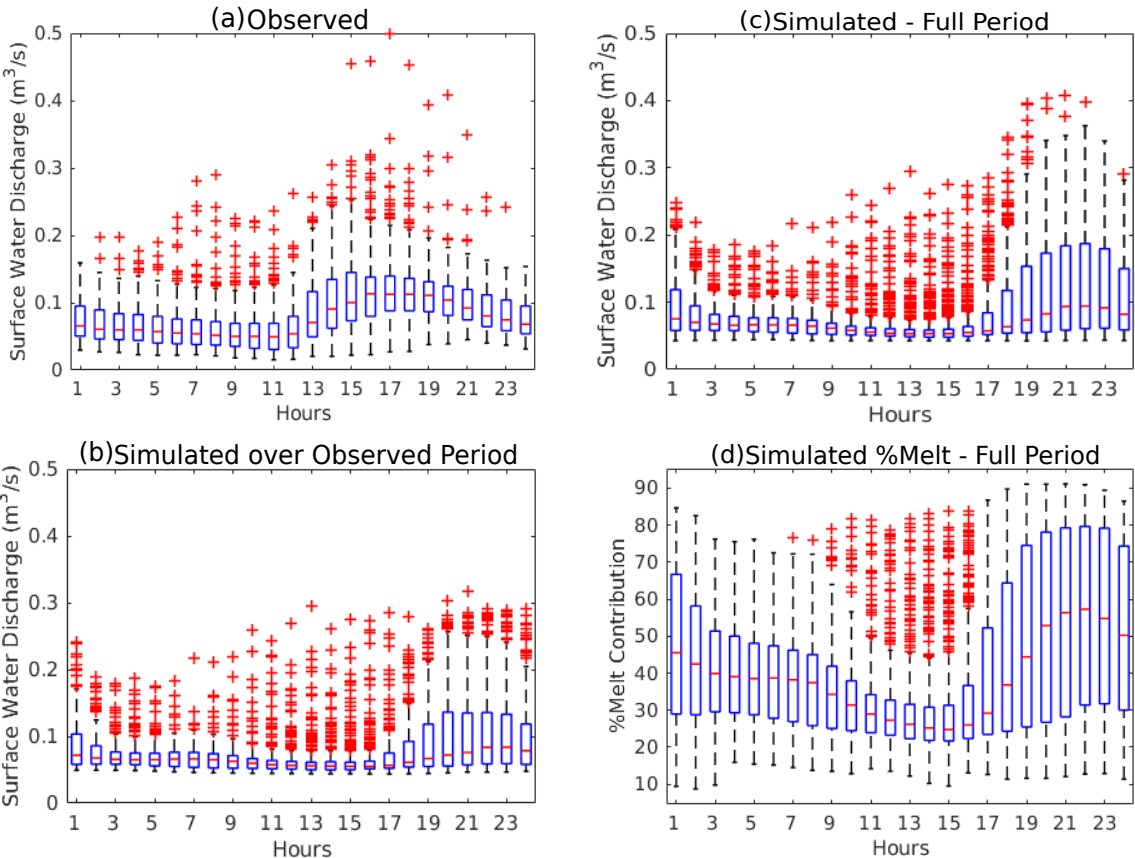

**Figure 7.** Box plots over local hour-of-day showing 25 – 75 percentiles with boxes and maximum and minimum with whiskers for **(a)** observed discharge over the 9-month observation period, **(b)** simulated discharge over the 9-month observation period, **(c)** simulated discharge over the 12-month simulation period, and **(d)** simulated percent glacier melt contribution to discharge over the 12-month period. Simulations do capture diurnal patterns, but with an underestimated magnitude and shifted peak. Simulated percent glacier melt contributions to discharge closely mirror the simulated diurnal fluctuations in discharge.

The estimate provided by equation (3) may not exactly correspond to the HBCM result for meltwater contribution for a number of reasons. First, the water balance impact of glacier melt conveyed in equation (3) may not equal the proportion of meltwater in a sample of discharge water if, for example, melt inputs facilitate more runoff of precipitation-sourced water. Also, while equation (3) aims to isolate glacier melt of pre-existing glacier ice, HBCM estimates could include snowmelt and melt of freshly accumulated ice, depending on the composition of the meltwater sample taken just below the glacier tongue. Lastly, any melt that infiltrates is considered as part of the groundwater rather than melt contribution in HBCM estimates, while equation (3) includes the effect of both surficial and groundwater contributions of meltwater to discharge.

These conceptual discrepancies complicate comparisons between the two methods, but it is possible to assess the June 20-25, 2016 HBCM result against the simulation (other HBCM periods fall outside the one-year model period). During June 20-25,

2016, the 45-64% estimate with HBCM is higher than the average simulated relative melt contribution of 29%, but it falls within the simulated hourly range of 13-70% over that time. Considering that samples were collected during the daytime when melt contributions were generally high, the results from these two approaches reasonably agree.

While process-driven temporal patterns were difficult to glean from the sparsely spaced and uncertain HBCM results, Flux-PIHM simulations in Figure 6(d) clearly indicate considerable variability in weekly glacier melt contributions of 25–61% over June 2015–June 2016. This range compares very closely with the 23–66% range bounded by the five HBCM estimates spanning January 2012 to February 2017 (indicated by the shading in Figure 6(d)), lending further confidence in the consistency between the watershed and mixing model results, despite the differences in temporal and other types of representation noted above. Hourly distributions of simulated glacier melt contribution in Figure 7(d) show an average diurnal range of 25-68%; given that the actual range is likely even broader due to underestimations by the temperature index model, diurnal fluctuations in relative melt contributions may be of similar or even greater magnitude than changes on weekly to monthly timescales.

Comparing Figures 6(c) and (d) reveals a remarkably strong correlation (0.89) between simulated weekly discharge and relative glacier melt contributions, indicating that major discharge peaks over multi-day timescales are melt-driven, a time-scale link that was masked in the time series analysis in Section 4.2 by the overwhelming diurnal signal between temperature and discharge. Figure 8(b) shows strong coherence over 30 to 80 day periods between simulated glacier melt production and relative glacier melt contributions to discharge, highlighting those timescales as the most prominent for direct glacier melt inputs beyond diurnal time scales. Although occurring at a longer multi-month timescale, these melt contributions to discharge nonetheless appear to happen mostly through fast surface runoff; the model showed most (86%) of the glacier melt contribution to discharge occurring through surface runoff processes, consistent with the hourly correlations found between observed discharge and temperature in Section 4.2. With the model, the influence of glacier melt production on discharge is clear during times such as early January 2016, when an uptick in discharge occurs despite low precipitation, due to warm temperatures and greater melt production and contribution (Figure 6).

Unexpectedly, the peak relative glacier melt contributions (Figure 6(d)) do not always align with glacier melt production patterns (Figure 6(a)) (e.g., in early February 2017, percent glacier melt contribution peaks just when temperature and glacier melt production dip), and their weekly correlation is not strong at a 95% confidence level (Figure S11(b)). Squared coherences in Figure 8(a) indicate that this is because relative glacier melt contribution also responds to precipitation, and this happens at weekly and multi-month timescales when correlations with glacier melt production are low (Figure 8(b)). In fact, the strong coherence between precipitation and glacier melt contribution around 6 to 14 day periods reveals that the high correlation between observed weekly precipitation and discharge in Section 4.3 relates to glacier melt contributions. Inspection of time periods such as mid-February and the start of April 2016 show that rainy periods can augment glacier melt contributions to discharge (Figure 6(b), (d)) . During those times, increases in relative glacier melt contributions to discharge coincided with precipitation events during local drops in glacier melt production, possibly because week-long precipitation creates antecedent moisture conditions that enhance the fraction of glacier meltwater that runs off over the surface. Surface runoff can be expected to contain mostly glacier meltwater or high-elevation precipitation, because runoff generally does not occur on low-elevation páramo soils except under intense precipitation (Sarmiento, 2000; Harden, 2004).

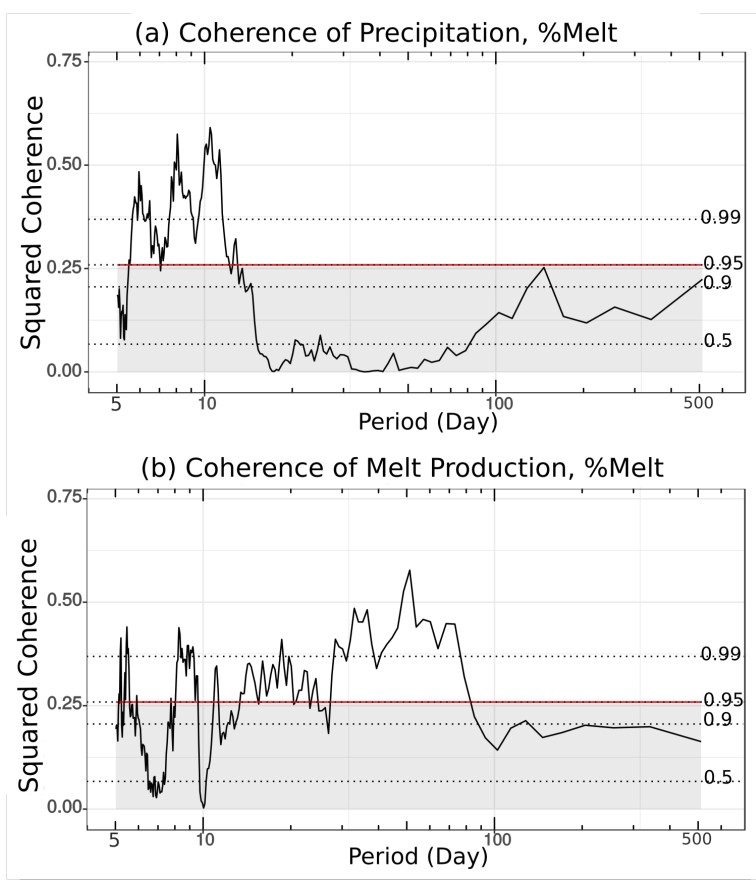

**Figure 8.** Magnitude squared coherences between simulated percent glacier melt contribution to discharge with **(a)** precipitation and **(b)** simulated glacier melt production. Calculations used daily average data to avoid known uncertainties in modeling diurnal fluctuations of glacier melt production. Precipitation and glacier melt production appear to be related to glacier melt contributions at complementary timescales.

At greater timescales, significant coherences between precipitation and simulated relative glacier melt contributions around 120+ days (Figure 8(a)) correspond to overall lowest relative glacier melt contributions during the dry El Niño period (December 2015-February 2016) and highest relative contributions during the two wet seasons (February-May and October-November), possibly due to a transfer of heat from rain to ice. These glacier melt contribution responses to bi-seasonal to interannual climatic patterns likely occur through slower groundwater pathways; the simulation with glacier melt produces 18% greater groundwater contributions to discharge than that without glacier melt (Figure 6(c)). This supports the hypothesis that some amount of glacier meltwater infiltrates and recharges to groundwater that then discharges farther downgradient to the stream channel. It should be noted, however, that the subsurface component of Flux-PIHM contains only a single shallow saturated zone, which mostly closely corresponds to a perched water table in the soil zone. This model implementation may not accurately represent additional deeper fractured bedrock aquifers systems, which could discharge groundwater to streams

over shorter, multi-month time scales (Andermann et al., 2012), in contrast to the relatively constant groundwater dynamics simulated in Figure 6(c).

Our results may be particular to the moist climate in the Gavilan Machay watershed, which supports constant groundwater discharge to the stream throughout the year and reflects compounded effects of melt inputs and rainfall conditions. However, although glacier melt intensifies discharge variability in this watershed, a baseline 25% glacier melt contribution throughout the simulation period indicates that a constant minimum level of glacier melt helps prevent episodes of even more extreme low flows during drought times, such as during El Niño conditions.

## 5   Summary and Conclusions

Although meltwater is typically credited with modulating stream discharge by buffering periods of low precipitation, we demonstrate using a combination of methods that relative meltwater contributions may drive nearly all the variability in discharge (correlation of 0.89) over a range of hourly to multi-year timescales while buffering only against extreme low discharge periods in a humid glacierized watershed in the Ecuadorian Andes. Hydrochemical mixing model results for five sampling periods spanning 2012-2017 showed the meltwater fraction in discharge may have varied over approximately 20–65%. Hydrologic model simulations over June 2015–June 2016 produced a nearly identical range for weekly contributions. The model also predicted a very similar average diurnal range, which likely provided a lower bound of actual variability based on hydroclimatic data. This multi-scale variability in melt contributions can be attributed to dynamic climate forcings that also contain a range of temporal patterns (Figure 9). We found a strong correlation between diurnal temperature and discharge changes that likely reflects melt production and supports the use of a temperature index melt model (Figure 9(a)). Although such a simple melt approach somewhat underestimated hourly fluctuation extremes with a lag, it led to reasonable weekly discharge predictions when implemented with a seasonally variable melt factor that possibly accounts for additional heat transfer from rainfall to ice during wet seasons. Spectral coherence analysis of the model results showed that not only were diurnal discharge patterns responding to melt production, but relative melt contribution and discharge variations over 30-80 day periods were controlled by extended glacier melt production periods that also contribute to discharge through surface runoff (Figure 9(c)).

Unexpectedly, model results showed that precipitation also boosted melt contributions to discharge, but on weekly and semi-annual timescales that complement the hourly and monthly timescales controlled by temperature and melt production. Weekly precipitation events likely generate antecedent wet conditions that facilitate greater amounts of meltwater runoff (Figure 9(b)), while longer-term precipitation patterns appear to drive slow changes in melt additions to groundwater (Figure 9(d)). Most (86%) melt contributions to discharge occurred through surface runoff in the model, but some meltwater recharged to groundwater, helping to support a relatively steady groundwater discharge to the stream that is about 18% greater with glacier melt than without glacier melt. As expected, strong El Niño conditions corresponded to some of the highest simulated melt inputs, but less easy to predict was that Gavilan Machay exhibited its lowest discharge during this time. Melt prevented streamflow from dropping below a baseline level during the warm and dry El Niño event, but discharge was much higher during wetter periods, in part because of the rainfall-enhanced melt contributions described above.

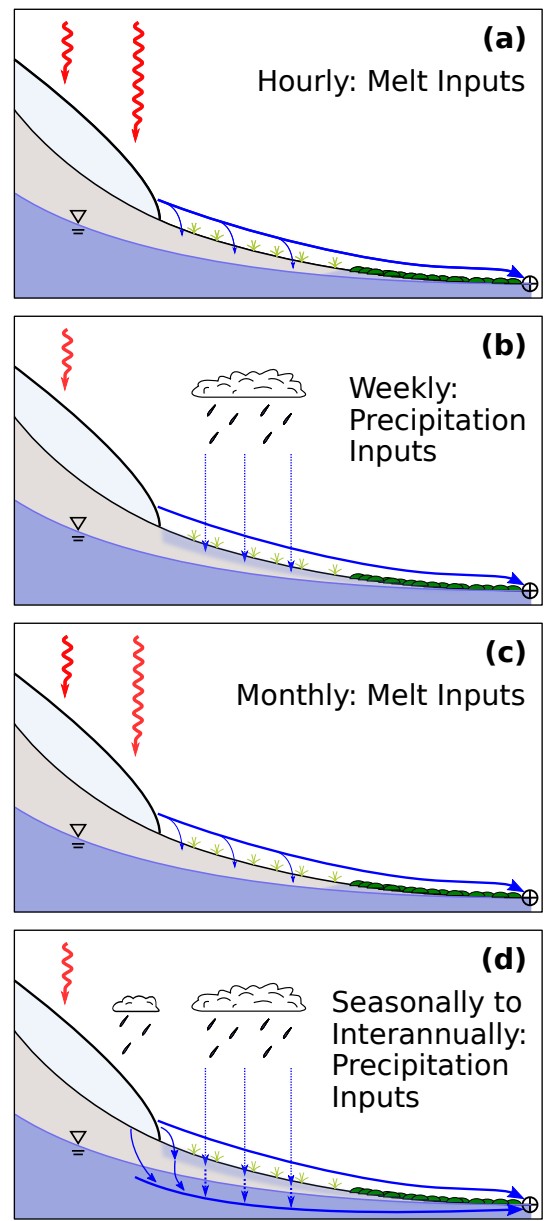

**Figure 9.** Relative melt contributions drive nearly all the variability in discharge in Gavilan Machay, mostly through surface runoff of glacial meltwater. What drives the variability in relative melt contributions to discharge? Our results show that this depends on the timescale. **(a)** Hourly timescale variability is controlled by radiation-driven (red arrows) melt production (light blue slab at upper left), which readily runs off overland (thick blue arrow) and eventually reaches the watershed's discharge point (circle with cross). **(b)** Weekly timescale variability is controlled by weekly precipitation events, which likely generate antecedent moisture conditions (light blue shading) that promote greater meltwater runoff. **(c)** Monthly timescale variability is driven by monthly trends in melt generation, which contributes to discharge mostly through surface runoff. **(d)** Seasonal to interannual variability is driven by long-term precipitation, which can enhance melt by transferring heat from rain (blue arrows right of glacier tongue), and augment subsurface melt contributions through increased groundwater flow (thick blue arrow below water table).

In glacierized watersheds, slower glacier melt contributions through groundwater are poorly constrained. Past studies have identified a component of meltwater in the groundwater (Favier et al., 2008; Lowry et al., 2010; Baraer et al., 2015; Minaya, 2016; Harrington et al., 2018), but to our knowledge, our work is the first to quantify this component. Generally, fractured young volcanic bedrock systems can support extensive groundwater (Tague et al., 2008; Frisbee et al., 2011; Markovich et al., 2016) and may contain similar meltwater fractions as found here. However, meltwater-groundwater interactions may be more ubiquitous. Prominent groundwater pathways have also been identified in fractured crystalline bedrock (Tague and Grant, 2009; Andermann et al., 2012; Pohl et al., 2015), morainic deposits (Favier et al., 2008; Minaya, 2016; Somers et al., 2016), and alpine meadow soils (Loheide et al., 2009; Lowry et al., 2010; Gordon et al., 2015). Even in settings with limited groundwater networks, talus slopes and rock glaciers can serve as localized areas of meltwater recharge (Clow et al., 2003; Baraer et al., 2015; Harrington et al., 2018). More arid settings than the sub-humid Gavilan Machay could have a higher proportion of glacier melt in groundwater due to less precipitation inputs, but our results also indicate that precipitation may serve to enhance meltwater contributions.

The multiscale temporal variability of relative melt contributions to discharge has important implications for how to determine the hydrologic role of glaciers in watersheds, as well as for water resource management in fast-changing glacierized systems. Care must be taken in the implementation and interpretation of commonly employed tracer analyses. Potentially large diurnal fluctuations make it imperative to collect samples over consistent times of day, and weekly to interannual variability complicate extrapolations from single synoptic sampling estimates. Recharge by glacier melt further confounds the interpretation of groundwater as a source entirely distinct from surficial meltwater. These uncertainties, along with additional errors caused by heterogeneous groundwater chemistry and the choice of sample locations, limit the ability of tracer-based analyses to constrain dynamic melt contributions to discharge. Model simulations provide ideal temporal resolution, but they suffer from their own disadvantages, such as intensive data input requirements and uncertainties in parameter calibration.

For water resources, weekly to multi-year melt contributions to discharge are of greater interest than hourly fluctuations. At those timescales of concern, rain events and wet periods can accentuate relative melt contributions to streamflow in humid glacierized systems. This signifies a bonus in water yield, but it also intensifies discharge variability over weekly and seasonal timescales, which can pose challenges if water storage infrastructure is unavailable. On an interannual basis, melt can augment discharge during warm and dry El Niño events, but glacierized watersheds will likely experience greatest melt contributions during wetter times, when they are under the combined effects of enhanced ablation (due to transfer of heat from rain to ice) and antecedent moist conditions (due to precipitation). Looking at downstream implications, the small (7.5 km$^2$) Gavilan Machay headwater catchment contributes a range of only 9 to 26% of the discharge to the Boca Toma diversion point, which corresponds to surficial meltwater making up a range of just 4 to 15% of the water to the irrigation system, based on mixing model estimates. However, La Frenierre (2014) showed that farming communities cannot afford to lose any of the water; already, the irrigation system consistently fails to deliver its current allocations. Furthermore, if groundwater at Gavilan Machay also contains meltwater, as our simulations suggest, the actual total amount of meltwater contribution could be even higher than that estimated for surficial runoff of meltwater by the mixing model. Additional downstream monitoring would enable further extensions of the watershed model to investigate how meltwater contributions and temporal discharge variabilities found in

Gavilan Machay propagate downgradient to successively larger watersheds, as non-glacier sourced groundwater contributions increase. Spatial patterns of groundwater contributions within Gavilan Machay reveal sharp increases where geologic features likely create localized discharge points (Figure S12). This indicates that extrapolations downstream will likely depend on geological conditions that control groundwater, in addition to watershed size and climate inputs.

In the future, should Reschreiter Glacier disappear completely, overall discharge in Gavilan Machay could decrease by up to about 50%, even if precipitation and temperature remain the same and relatively constant groundwater flow continues. The exact decrease will depend on how much of our melt estimate may include contributions by snowmelt and melt from freshly accumulated ice that will persist as discharge sources. Overall, our findings suggest that in response to glacier loss in a warming climate, glacierized watersheds in the humid inner tropics may eventually experience steadier discharge, but

potentially at significantly decreased rates.

*Code and data availability.* The version of Flux-PIHM used for this paper is available on GitHub, at https://github.com/PSUmodeling/MM-PIHM. The PIHMgis software used to create input files is available at http://www.pihm.psu.edu/pihmgis_downloads.html. The corresponding author may be contacted to access field data collected and used in this work.

*Author contributions.* Saberi implemented the time series analysis and hydrologic modeling, assisted with the mixing model analysis, and
contributed to writing the manuscript. McLaughlin executed the field work in 2016-2017, implemented the mixing model analysis, and contributed to writing the manuscript. Ng conceived of the project together with La Frenierre, oversaw the analysis, and contributed to writing the manuscript. La Frenierre also established the study site and led the fieldwork. Wickert provided field instrumentation for 2015-2017 and contributed to fieldwork. Baraer provided the HBCM mixing model software and guided its implementation. Zhi and Li assisted with the Flux-PIHM hydrologic model implementation. Mark supported establishment of the field site and supervised initial work.

*Competing interests.* The authors declare that they have no conflict of interest.

*Acknowledgements.* Funding from NSF (EAR-1759071) helped support this work. McLaughlin received travel support from University of Minnesota's Walter H. Judd Fellowship. Ng and Wickert received financial support from University of Minnesota. La Frenierre received financial support from the National Science Foundation's Doctoral Dissertation Research Improvement Grant (#1103235); the Fulbright Commission of Ecuador, the Geological Society of America, and Gustavus Adolphus College. The authors would like to acknowledge field
assistance from Chad Sandell, Casey Decker, Helen Thompson, and Abigail Michels; lab support from Jeff Jeremiason, Chris Harmes, and Scott Alexander; and land-cover mapping assistance from Josh Zoellmer.

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
