# Peer review of "Multi-scale temporal variability in meltwater contributions in a tropical glacierized watershed"

_Hydrology and Earth System Sciences, 2018_

## Referee Comment (RC1) · Anonymous Referee #1 · 12 Jun 2018

This paper presents a detailed multi-method assessment of glacier melt and groundwater contribution to runoff for a small catchment in the tropics. The authors find significant contributions of melting to overall runoff using tracer studies, time series analysis and hydrological modelling. They also show that melt water can be a substantial contributor to groundwater discharge.

This is an excellent study that presents a thorough analysis of field data and modelling leading to interesting conclusions. The manuscript is very well written, and methods and results are clearly described. The findings are also nicely presented. Overall, I absolutely recommend this paper for publication in HESS after some – mostly minor – issues have been resolved (see below).

More substantive comments:

[Figure]

- Page 5, line 11: Is there an estimate how important glacier-derived runoff is for the larger catchment? A high importance (irrigation system) is implied here, but how does the glacier runoff volume relate to larger-scale effective precipitation? Given that the absolute runoff amounts in the Gavilan Machay basin are really small (in the order of 0.1 m3/s) I doubt that this water (despite of originating from the headwaters) has a major significance lower downstream. This is also supported by the statement of page 12, line 6. The glaciers' importance for water resources in the region might need to be better put into context.

- Page 9, line 23: The authors use a model that computes snow melt based on the energy balance. It is surprising to me that they nevertheless decided to implement an empirical, strongly simplified model for ice melt. This seems to be an unnecessary and also unphysical combination of approaches. Later, it is stated that a temperature index model is the only feasible approach given the limited data availability. However, if data are available to force an energy balance model for snow, it should also be applicable to glacier ice (just having a different albedo and surface roughness). More argumentation is required here, and possibly more insight into the energy balance scheme of Flux-PIHM.

- Page 9, Eq. 1: Given that relatively large parts (those experiencing the highest melt rates) of the glaciers are covered by supraglacial debris, I wonder how the model distinguishes between ice melting over these regions in comparison to clean ice.

- Figure 4: In my print-out (but not in the online pdf version!) there are ugly black squares around panels c and d, mostly covering the axis text. Please carefully check the figure data. Obviously, these issues only arise for particular printer drivers but make the figure almost unreadable. The same observation has also been made for Figure 8 (black squares left of the glacier snout in all panels).

- Page 20, line 12: Tackling the problem using different complementary approaches is highly beneficial. However, after reading the results section I somewhat missed a

synthesis (figure) of the findings from the three different methodologies. For example, Fig. 3 and Fig 5 c/d could be combined to permit a direct comparison of findings based on tracers and based on the hydrological model which might also be helpful in discussing drawbacks and potentials of the individual methods.

Additional detailed comments: - Page 2, line 11: normally, references are ordered with the year of appearance but not here.

- Page 3, line 5: please shortly mention the physical reason (energy balance) why higher humidity leads to more ablation – this might not be immediately clear to the reader.

- Page 5, line 19: Are there observations of recent glacier retreat in this region? Just to round up the story.

- Page 5, line 27: precipitation gradients were determined with stations at 3900 and 4500 m a.s.l., respectively. Will this elevation difference be enough to capture / estimate precipitation over the higher reaches of Chimborazo, i.e. between 5000 to 6200 m a.s.l.?

- Page 5, line 28: It is a drawback for melt model validation that the ablation stakes are only installed over a very limited elevation range (i.e. not permitting to capture elevation gradients in glacier melt), and – as it seems – only over the debris-covered parts of the glaciers. This should be stated.

- Figure 7: I like the analysis of the coherences and it allows interesting conclusions to be drawn. However, it would be helpful if the term "coherence" would be better introduced, making it clearer how it was computed and what it potentially shows.

- Page 19, line 12: Highly interesting finding. In how far could these 18% meltwater contribution to groundwater runoff be generalized to other catchments (different sizes, geology etc.)? Have there been other studies coming up with similar estimates or is this the first time this has been quantified? May be something for the conclusion section.

- Page 22, line 22: I do not agree that runoff after glacier disappearance decreases by the current amount of melt contribution. As much as I understand, melt computed by the model includes both ice and snow melt. Whereas glacier ice melt is zero after the glacier has disappeared, snow melt is likely to remain a significant component of runoff or would be replaced by liquid precipitation in the case that the zero degree line remains above the top of Chimborazo all the time. Therefore, I would expect a significantly smaller runoff reduction for the catchment in the far future than implied here.

---

## Referee Comment (RC2) · Anonymous Referee #2 · 17 Oct 2018

This paper is an excellent, in-depth exploration of multiple methods to constrain the role of meltwater in downstream hydrology, granted at a very small scale. The innovative contribution is the use of different time scales to demonstrate the interplay between melt regimes, precipitation patterns, and groundwater dynamics. There is tremendous opportunity to expand the relevance of this work in the future by applying a similar suite of methods to data from further downstream, or nested catchments.

A few overarching issues that should be more clearly addressed in the discussion/conclusions:

1. The 7.5 kmˆ2 basin study area has offered valuable insights because of the data collection and monitoring that can be done at this scale, but it is important to acknowledge how your insights and results may translate downstream, given that your interpre-

tations of the hydrology and implications of glacier recessions on discharge are being presented for an area in immediate proximity to the glacier terminus.

2. The differentiation, or lack thereof, between snow and glacier melt should be more explicitly discussed. How big a role does snow (melt) play in the catchment, and what data to you have that informs this? To explore this, and relevant to many of your interpretations, a cursory estimate of the precipitation partition in the catchment could be interesting - what percent of precip falls as snow vs. rain based on your temperature and precip data? Given that information and your discharge measurements, do you have a sense of relative contribution of snowmelt vs. glacier ice melt, or even how much of the discharge from the glacier terminus could also be liquid precip routed through that pour point?

Specific comments:

P3 L23-24: which 4 major river systems?

P5 L5: Do you know if historically any other glaciers generated perennial surface discharge?

P6 L10-12: Lack of any rainy/wet season samples is a limitation.

Section 3.2.2: Having run same analytes in different labs in different years potentially introduces error or uncertainty. How confident are you in comparing different lab results? E.g. were detection limits the same, were any lab comparisons done?

Figure 2: 2(a) and 2(c) read like results.

P8 L20: grammar – 'is be unique'

P10 L10: how were T, P, and RH interpolated?

P11 L4-7: unclear here how you ultimately selected tracers for the mixing model. E.g. were thresholds applied to bivariate plots?

P11 L13-17: Any hypothesis on why groundwater discharge was so low in Feb 2017? Are there temps or precip events that inform this anomaly?

Figure 4 caption, line 4: "corresponding to the"

P14 L13-15: how do these melt factors compare to the literature?

P14 L24: reference for historic geodetic mass balance estimates?

P14 L30: missing close parentheses - "full details)."

Figure 5(d): y-axis label typo – "Contribution"

Figure 5 caption: clarification on "(a) average air temp below ELA (5050 m.a.s.l.) and over glaciers and simulated melt inputs" – does this mean T is averaged over ablation zone plus snow covered area?

Figure 5 caption, L4: 'distribution' should be 'contribution'

P20 L10-11: what you suggest here is a buffer against lower extreme low flows during drought times, which somewhat contradicts your repeated assertion (e.g. P20 L14) that melt does not necessarily provide the buffer often credited to it. Reconciling these, perhaps with a clear acknowledgement in the conclusions that the buffer does exist for extreme low flow scenarios, but the modulating effect in other flow scenarios may not be as strong of a control on streamflow as other studies suggest.

P20 L26-27: these longer periods controlled by melt inputs are via infiltration and groundwater recharge, right?

Figure 8 caption, lines 1-2: reference here to glacial meltwater is misleading, since what you've characterized is glacier outflow that is a combo of ice and snow melt, right?

P22 L9: "Recharge by meltwater"

P22 L22-23: Unclear what justifies the assertion that discharge could be reduced by

half. Equilibrium discharge with glacier melt contributions and equilibrium discharge post-glaciers should be the same if precip is the same, barring other changes (e.g. increased ET). The peak water period in the middle is a different story, but this claim seems unsupported.

P22 L24-25 Related to the previous comment and as mentioned at the beginning, the other huge caveat is that you are looking at a point 2 km from a glacier terminus, so results absolutely cannot be implied to inform understanding of vulnerability of water resources. Extrapolating further downstream is a logical next step and I think expanding your methods downstream would be an incredibly valuable contribution to this understanding!!!

―――――――――――――――――――

---

## Author Comment (AC1) · 1 Nov 2018

[letterpaper, 11pt]article [english]babel inputenc amsmath graphicx [left=1in,top=1in,right=1in,bottom=1in]geometry [hyphens]url [super, comma, sortcompress]natbib tcolorbox wrapfig nicefrac [font=small,labelfont=bf, belowskip=0pt]caption enumitem tabu [normalem]ulem breakcites titlesec comment silence latexText page setspace parskip amsmath makecell booktabs,tabularx x>X footnote adjustbox caption [table]skip=35pt multicol color,soul xcolor

[Figure]

**1 Response to Reviewer 1**

We would like to thank the reviewer for their time to review our paper. In this response, we have addressed all the reviewer's comments by providing clarifications and indicating how we will edit the manuscript. The reviewer's comments are copied here with italic font style.

*This paper presents a detailed multi-method assessment of glacier melt and groundwater contribution to runoff for a small catchment in the tropics. The authors find significant contributions of melting to overall runoff using tracer studies, time series analysis and hydrological modelling. They also show that melt water can be a substantial contributor to groundwater discharge. This is an excellent study that presents a thorough analysis of field data and modelling leading to interesting conclusions. The manuscript is very well written, and methods and results are clearly described. The findings are also nicely presented. Overall, I absolutely recommend this paper for publication in HESS after some – mostly minor – issues have been resolved (see below).*

We are encouraged by the reviewer's positive comments and will carefully address all issues raised.

*More substantive comments:*

*Page 5, line 11: Is there an estimate how important glacier-derived runoff is for the larger catchment? A high importance (irrigation system) is implied here, but how does the glacier runoff volume relate to larger-scale effective precipitation? Given that the absolute runoff amounts in the Gavilan Machay basin are really small (in the order of 0.1 $m^3/s$) I doubt that this water (despite of originating from the headwaters) has a major significance lower downstream. This is also supported by the statement of page 12, line 6. The glaciers' importance for water resources in the region might need to be better put into context.*

When extended downgradient to the Boca Toma diversion point, our mixing model
analysis with HBCM predicts that surficicial runoff of meltwater contributes a range of 4-15% of the discharge to the irrigation system over 2012-2017, with the rest supplied by groundwater. While this melt contribution indeed seems to comprise a minor proportion, earlier investigation by La Frenierre (2014) on downstream water usage showed that farming communities cannot afford to lose any of the water; already, the irrigation system consistently fails to deliver its current allocations. Furthermore, if groundwater at Gavilan Machay also contains meltwater, as our simulations suggest, the actual total amount of meltwater contribution could be even higher than the 4-15% estimated for surficial runoff of meltwater. We will add this discussion in our revised manuscript. A lack of model input data outside of the Gavilan Machay sub-catchment prevented further extension of the model to Boca Toma.

Reference:

La Frenierre, J., 2014, "Assessing the Hydrologic Implications of Glacier Recession and the Potential for Water Resources Vulnerabilities in Volcán Chimborazo, Ecuador", PhD Dissertation, Ohio State University, 2014.

*Page 9, line 23: The authors use a model that computes snow melt based on the energy balance. It is surprising to me that they nevertheless decided to implement an empirical, strongly simplified model for ice melt. This seems to be an unnecessary and also unphysical combination of approaches. Later, it is stated that a temperature index model is the only feasible approach given the limited data availability. However, if data are available to force an energy balance model for snow, it should also be applicable to glacier ice (just having a different albedo and surface roughness). More argumentation is required here, and possibly more insight into the energy balance scheme of Flux-PIHM.*

We do not use energy balance calculations for glacier melt for two reasons. First, energy balance calculations of glacier melt would have to be coupled with the other energy balance calculations already in the Flux-PIHM model (for snow-melt, ET, sensible

heat flux, and ground heat flux) because of both its role in the partitioning of incoming net radiation and its effect on surface temperature. However, adding this to Flux-PIHM requires intensive source-code modifications that are beyond the scope of this study. Second, an alternative approach of an approximate, uncoupled energy balance calculation of glacier-melt would be complicated by the lack of radiation input measurements in the study watershed. We currently use GLDAS data with Flux-PIHM for its energy balance calculations, but there is substantial uncertainty in applying the coarse-scale GLDAS radiation values over the steep mountainous watershed. Because of these difficulties, we chose to invoke the simpler temperature-index model and focused on constraining glacier-melt amounts based on discharge observations at the watershed outlet. We note that using coarse-scale GLDAS does introduce uncertainty into the current Flux-PIHM energy balance calculations, including for snowmelt. However, even without partitioning some of the incoming radiation for glacier melt in the model, our simulated snowmelt is a relatively small contribution of the total meltwater (15%), suggesting that precipitation limitations may make snowmelt calculations less sensitive to uncertainties in radiation inputs. We will better explain our choice of the temperature index model in the revised manuscript and acknowledge the corresponding uncertainties.

*Page 9, Eq. 1: Given that relatively large parts (those experiencing the highest melt rates) of the glaciers are covered by supraglacial debris, I wonder how the model distinguishes between ice melting over these regions in comparison to clean ice.*

We reported measurements of a slower ablation rate (0.54 to 0.87 m/yr) for the insulated debris-covered ice compared to a faster rate (3.4 m/yr) for the clean ice (p. 14, Lines 23-25), which indeed support debris-dependent melt conditions. However, these were only a handful of ablation measurements over different time periods, which were not sufficient to constrain separate melt factors for debris-covered and clean ice. We thus elect to use an effective melt factor over all glacierized areas (below the equilibrium line altitude) to model the bulk rate.

*Figure 4: In my print-out (but not in the online pdf version!) there are ugly black squares around panels c and d, mostly covering the axis text. Please carefully check the figure data. Obviously, these issues only arise for particular printer drivers but make the figure almost unreadable. The same observation has also been made for Figure 8 (black squares left of the glacier snout in all panels).*

We will update these figures in the manuscript. Thank you for pointing this out.

*Page 20, line 12: Tackling the problem using different complementary approaches is highly beneficial. However, after reading the results section I somewhat missed a synthesis (figure) of the findings from the three different methodologies. For example, Fig. 3 and Fig 5 c/d could be combined to permit a direct comparison of findings based on tracers and based on the hydrological model which might also be helpful in discussing drawbacks and potentials of the individual methods.*

Thank you for the suggestion. We will update Fig. 5d (showing model results) by shading the interval of % Melt Contribution estimated with the mixing model (from Fig. 3a) in order to facilitate comparisons between methods. Adding the discharge estimates from the mixing model (from Fig. 3b) to Fig. 5c would likely make the plot too busy, since it already has 5 different lines. The discharge information is summarized in Fig. 5d, so we think that adding the mixing model results to Fig. 5d should suffice. We will also edit the caption/text to remind the reader that the results represented by the mixing model somewhat differ from that of the watershed model, because the mixing model results included 5 discrete sampling times that were distinct from the simulation period, and the mixing model does not consider meltwater in groundwater.

*Additional detailed comments:*

*Page 2, line 11: normally, references are ordered with the year of appearance but not here.*

Thank you for pointing this out. We will update the order of references in the manuscript.

*Page 3, line 5: please shortly mention the physical reason (energy balance) why higher humidity leads to more ablation – this might not be immediately clear to the reader.*

We will add the following explanation to the manuscript: "Harpold and Brooks (2018) showed that increasing humidity enhances ablation rate by increasing net longwave radiation and condensation."

*Page 5, line 19: Are there observations of recent glacier retreat in this region? Just to round up the story.*

We will revise the following sentences by adding the underlined explanation to page 5, line 1-4: "Records since 1980 indicate that, consistent with the rest of the tropical Andes, temperatures have warmed 0.11°$C$/decade around Volcń Chimborazo (Vuille et al., 2008; La Frenierre and Mark, 2017). This likely caused a 21% reduction in ice surface area and 180m increase in the mean minimum elevation of clean ice between 1986 and 2013 (La Frenierre and Mark, 2017)".

*Page 5, line 27: precipitation gradients were determined with stations at 3900 and 4500 m a.s.l., respectively. Will this elevation difference be enough to capture / estimate precipitation over the higher reaches of Chimborazo, i.e. between 5000 to 6200 m a.s.l.?*

Previous research in a glacierized mountainous watershed by Wang et al. (2016) found that the elevation-precipitation relationship is piecewise linear, with precipitation increasing with altitude below the elevation of maximum precipitation (EMP) and decreasing with altitude above the EMP. Such results support our application of a negative linear lapse rate calculated from our two stations – both located in the lower part of the watershed – to the higher elevation portions of the watershed. However, we should and will explicitly acknowledge that this assumes the EMP to be located below our watershed, which could lead to errors in the precipitation if the EMP is actually within the watershed above the lowest weather station. We will point out the need for denser monitoring to better constrain the EMP and precipitation lapse rate.

Reference:

Wang, X., Sun, L., Zhang, Y., and Luo, Y. (2016). "Rationalization of altitudinal precipitation profiles in a data-scarce glacierized watershed simulation in the Karakoram." Water, 8(5), 186.

*Page 5, line 28: It is a drawback for melt model validation that the ablation stakes are only installed over a very limited elevation range (i.e. not permitting to capture elevation gradients in glacier melt), and – as it seems – only over the debris-covered parts of the glaciers. This should be stated.*

The ablation stakes were installed in clean ice. We will make sure to explain this in the Methods section (it is currently only mentioned later in the Results section, p. 14 Line 24) and acknowledge that the ablation stakes do not represent debris-covered ice, which was too difficult to drill into. Recognizing the limited representation of the ablation stakes, we do not directly use their measurements in the model but instead only use them as a high-end point of comparison for our calibrated glacier-melt model. Later in Section 4.3.1 Calibration Results, we explain that our calibrated average glacier melt rate (below the equilibrium line altitude) is lower than the ablation stake measurements in faster-melting clean ice and higher than the mass balance measurements for the slower-melting debris-covered ice (p. 14, lines 21-25).

*Figure 7: I like the analysis of the coherences and it allows interesting conclusions to be drawn. However, it would be helpful if the term "coherence" would be better introduced, making it clearer how it was computed and what it potentially shows.*

We present results for magnitude squared coherence (MSC), which can be thought of as the square of the correlation (between 0 and 1) between two variables at a certain frequency. Thus, coherencies between precipitation and discharge and between temperature and discharge indicate how strongly each of the climatic signals relate to discharge at a certain time scale. Looking at different time scales helps to distinguish whether these relationships may occur through fast surficial processes or slower subsurface processes, and whether discharge is more sensitive to certain climate forcings at particular time frequencies. MSC is defined as:

$$C_{xy} = \frac{|S_{xy}(f)|^2}{S_{xx}(f)S_{yy}(f)} \tag{1}$$

where, $S_{xx}(f)$ and $S_{yy}(f)$ are auto-spectral densities of variables $x$ and $y$, respectively, and $S_{xy}(f)$ is the cross spectral density of $x$ and $y$. This explanation will be added to the manuscript.

*Page 19, line 12: Highly interesting finding. In how far could these 18% meltwater contribution to groundwater runoff be generalized to other catchments (different sizes, geology etc.)? Have there been other studies coming up with similar estimates or is this the first time this has been quantified? May be something for the conclusion section.*

Past studies have examined the overall role of groundwater in glacierized watersheds and have found it to contribute up to 80% of total stream discharge (Clow et al., 2003; Liu et al., 2004; Huth et al., 2004; Hood et al., 2006; Baraer et al., 2009; Andermann et al., 2012; Baraer et al., 2015; Pohl et al., 2015; Somers et al., 2016; Harrington et al., 2018). A smaller number of studies have also identified a component of meltwater in the groundwater (Favier et al., 2008; Lowry et al., 2010; Minaya, 2016; Baraer et al., 2015; Harrington et al., 2018), but to our knowledge, our work is the first to quantify this component. Generalizing our results to other glacierized watersheds depends on a number of geologic and climatic factors. The importance of meltwater contributions to streamflow through groundwater depends first on the presence of groundwater pathways. These typically are most prominent with the presence of fractures in young volcanic bedrock (Tague et al., 2008; Frisbee et al., 2011; Markovich et al., 2016) – like Chimborazo – and sometimes even crystalline bedrock (Tague et al., 2009; Andermann et al., 2012; Pohl et al., 2015). Morainic deposits (Favier et al., 2008, Minaya, 2016, Somers et al., 2016) and alpine meadow soils (Loheide et al., 2009; Lowry et al., 2010;

Gordon et al., 2015) have also proved to be effective groundwater units below glaciers and snowpacks. Even in settings that may have limited groundwater networks extending throughout the watershed, talus slopes can serve as localized areas of meltwater recharge (Clow et al., 2003; Baraer et al. 2015; Harrington et al., 2018). In the groundwater, the proportion of precipitation versus meltwater depends on watershed size and climate. Well-established discharge-watershed area relationships for non-glacierized watersheds (Dunne and Leopold, 1978) lead to predictions of increased precipitation contribution in larger watersheds (with similar glacierized areas). More arid settings may be expected to have a higher proportion of glacier-melt due to overall less precipitation inputs to the watershed, although our results indicate a possible interaction between glacial melt contributions and precipitation, where rainfall boosts melt contributions through both the transfer of heat to glaciers and through antecedent moisture conditions that facilitate meltwater recharge.

We thank the reviewer for this comment, which prompts us to better highlight our new contribution and its potential implications elsewhere.

New References (other references are in the original reference list for the manuscript):

Dunne, T., and Leopold, L. B. (1978). Water in environmental planning. Macmillan.

Frisbee, M. D., Phillips, F. M., Campbell, A. R., Liu, F., and Sanchez, S. A. (2011). "Streamflow generation in a large, alpine watershed in the southern Rocky Mountains of Colorado: Is streamflow generation simply the aggregation of hillslope runoff responses?." Water Resources Research, 47(6).

Harrington, J. S., Mozil, A., Hayashi, M., and Bentley, L. R. (2018). "Groundwater flow and storage processes in an inactive rock glacier." Hydrological Processes. Hood, J. L., Roy, J. W., and Hayashi, M. (2006). Importance of groundwater in the water balance of an alpine headwater lake. Geophysical Research Letters, 33(13).

Huth, A. K., Leydecker, A., Sickman, J. O., and Bales, R. C. (2004). "A two‐com-

ponent hydrograph separation for three high‐elevation catchments in the Sierra Nevada, California." Hydrological Processes, 18(9), 1721-1733.

Pohl, E., Knoche, M., Gloaguen, R., Andermann, C., and Krause, P. (2015). "Sensitivity analysis and implications for surface processes from a hydrological modelling approach in the Gunt catchment, high Pamir Mountains." Earth Surface Dynamics, 3(3), 333-362.

Somers, L. D., Gordon, R. P., McKenzie, J. M., Lautz, L. K., Wigmore, O., Glose, A., ... and Condom, T. (2016). "Quantifying groundwater–surface water interactions in a proglacial valley, Cordillera Blanca, Peru." Hydrological Processes, 30(17), 2915-2929.

*Page 22, line 22: I do not agree that runoff after glacier disappearance decreases by the current amount of melt contribution. As much as I understand, melt computed by the model includes both ice and snow melt. Whereas glacier ice melt is zero after the glacier has disappeared, snow melt is likely to remain a significant component of runoff or would be replaced by liquid precipitation in the case that the zero degree line remains above the top of Chimborazo all the time. Therefore, I would expect a significantly smaller runoff reduction for the catchment in the far future than implied here.*

The reviewer's comment prompts us to make one clarification and also qualify our statement about the runoff change after the glaciers disappear. First, we clarify that the model scenario we called "No Melt" should have been called "No glacier melt", and the scenario we called "With melt" should have been called "With glacier melt" - we will correct this naming scheme in the manuscript. Both model scenarios include the same snowmelt amount, because they use the same meteorological inputs to Flux-PIHM. Flux-PIHM simulates snowmelt based on precipitation and temperature inputs, while glacier melt is simulated externally (using the temperature-index model) and then added as another water source to Flux-PIHM. Thus, our calculation of change between the two scenarios isolates the effect of having glacier melt versus no glacier melt.

Although by design our simulation scenarios aim to separate out glacier-melt and

snowmelt contributions, we acknowledge that we lack constraints on their individual amounts, which means that our calibrated glacier-melt contribution could incorporate precipitation-sourced meltwater not fully accounted for in Flux-PIHM's snowmelt scheme. Further, mixing model estimates of meltwater contribution uses meltwater from the glacier tongue, which may include snowmelt and melt of freshly accumulated ice. Thus, we will qualify our statement about the decrease in future runoff post-glaciers (under the same precipitation conditions): we will say that the estimated amount of current meltwater provides an upper limit, and the reduction could be less depending on snowmelt.

---

## Author Comment (AC2) · 1 Nov 2018

[letterpaper, 11pt]article [english]babel inputenc amsmath graphicx [left=1in,top=1in,right=1in,bottom=1in]geometry [hyphens]url [super, comma, sortcompress]natbib tcolorbox wrapfig nicefrac [font=small,labelfont=bf, belowskip=0pt]caption enumitem tabu [normalem]ulem breakcites titlesec comment silence latexText page setspace parskip amsmath makecell booktabs,tabularx x>X footnote adjustbox caption [table]skip=35pt multicol color,soul xcolor

**1 Response to Reviewer 2**

We would like to thank the reviewer for their time to review our paper. In this response, we have addressed all the reviewer's comments by providing clarifications and indicating how we will edit the manuscript. The reviewer's comments are copied here with italic font style.

*This paper is an excellent, in-depth exploration of multiple methods to constrain the role of meltwater in downstream hydrology, granted at a very small scale. The innovative contribution is the use of different time scales to demonstrate the interplay between melt regimes, precipitation patterns, and groundwater dynamics. There is tremendous opportunity to expand the relevance of this work in the future by applying a similar suite of methods to data from further downstream, or nested catchments.*

We are encouraged by the reviewer's positive comments and will carefully address all the issues raised.

*A few overarching issues that should be more clearly addressed in the discussion/conclusions:*

*1. The 7.5 $km^2$ basin study area has offered valuable insights because of the data collection and monitoring that can be done at this scale, but it is important to acknowledge how your insights and results may translate downstream, given that your interpretations of the hydrology and implications of glacier recessions on discharge are being presented for an area in immediate proximity to the glacier terminus.*

We will discuss potential downstream implications of our work through two edits to the manuscript.

First, we will explain that we did extend our mixing model analysis downstream of Gavilan Machay to the Boca Toma diversion point for an irrigation system. We found that the surficial glacial meltwater contribution dropped to about 4-15% of the discharge at

Boca Toma. Although this amount of meltwater appears to comprise a minor proportion of discharge, an earlier investigation by La Frenierre (2014) on downstream water usage showed that farming communities cannot afford to lose any of the water. Already, the irrigation system consistently fails to deliver its current full allocations. Furthermore, if groundwater at Gavilan Machay contains glacial meltwater, as our simulations suggest, the actual total amount of meltwater contribution could be even higher than the 4-15% estimated for surficial meltwater. A lack of model input data outside of the Gavilan Machay sub-catchment prevented further extension of the model to Boca Toma.

Second, we will discuss the potential outcome of extending measurements and the model implementation over nested watersheds covering successively larger downstream areas. We expect that a characteristic relationship may emerge between watershed size and meltwater discharge, similar to the exponential relationship found between relative groundwater (versus meltwater) contribution and glacierized area in the multi-watershed study in the Cordillera Blanca of Peru by Baraer et al. (2015). Our results on Volcán Chimborazo did not match their exponential prediction, indicating that the characteristic curves depend on climate and geologic setting. In fact, looking within our small study watershed, estimates of groundwater contribution over the stream network reveal a nonlinear relationship with subcatchment area that contains sharp increases where geologic features likely create localized discharge points (we will add the figure to the Supplementary Information). This indicates that extrapolations downstream depends on geological conditions that control groundwater.

Reference:

La Frenierre, J., 2014, “Assessing the Hydrologic Implications of Glacier Recession and the Potential for Water Resources Vulnerabilities in Volcán Chimborazo, Ecuador”, PhD Dissertation, Ohio State University, 2014.

*2. The differentiation, or lack thereof, between snow and glacier melt should be more explicitly discussed. How big a role does snow (melt) play in the catchment, and what*

*data to you have that informs this? To explore this, and relevant to many of your interpretations, a cursory estimate of the precipitation partition in the catchment could be interesting - what percent of precip falls as snow vs. rain based on your temperature and precip data? Given that information and your discharge measurements, do you have a sense of relative contribution of snowmelt vs. glacier ice melt, or even how much of the discharge from the glacier terminus could also be liquid precip routed through that pour point?*

We will edit the manuscript to clarify the distinction between snow and glacier melt in both the mixing model analysis and watershed model simulations.

For the mixing model, the meltwater end-member was represented using the hydrochemistry of a water sample collected just below the glacier tongue. Even during our dry season sampling, some of the meltwater could include melting of snow from the previous seasons, and so our estimates of meltwater contribution using the mixing model approach would include both glacier and snowmelt.

In our watershed model simulations, we aimed to distinguish glacier from snowmelt, but we realized from the reviewer's comment that we did not explain this clearly in the text. When we referred to the "With melt" and "No melt" model scenarios, we meant "With glacier melt" and "No glacier melt", respectively, because the two scenarios actually have the same amount of snowmelt; the only difference is that the former has an added glacier melt amount of water - determined with the temperature index model calibrated to discharge measurements. Apart from this glacier melt amount, both scenarios include the same meteorological inputs and thus same snowmelt amount. Through this approach, the melt contribution that we determined by differencing the two scenarios should only represent glacier melt. We will clarify this.

The reviewer does make the good suggestion that we should discuss the relative snowmelt and glacier melt amounts in the model. The model Flux-PIHM simulates the partitioning of precipitation inputs between rain and snow based on air temperature. Over the 2016 period, the model predicted that about 12% of precipitation falls as snow in the watershed. The simulated snowmelt amount is 15% of the total meltwater. On average, snow melt contributes 8% of stream discharge, while glacial melt contributes 52%.

We acknowledge that although we aim to distinguish between glacier melt and snowmelt in the model, there is uncertainty in the partitioning due to a lack of separate data constraints. Our snowmelt simulation is highly sensitive to any errors in the lapse-rate based spatial extrapolation of precipitation and temperature in the watershed, and so it is possible that our calibrated glacier melt includes some amount of snowmelt that is not captured in the Flux-PIHM simulation. We will edit the text to acknowledge this.

*Specific comments:*

*P3 L23-24: which 4 major river systems?*

The four major river systems are: the Río Mocha (NE flank), Río Colorado (NW flank), Río Guano (SE flank), and Río Chimborazo (SW flank). The names of the watersheds will be added to the manuscript.

*P5 L5: Do you know if historically any other glaciers generated perennial surface discharge?*

There are no historical streamflow data for any of Chimborazo's glacierized watersheds that are close enough to the mountain to be able to discern a glacier melt signature. Discussions with local water users by La Frenierre (2014) did not yield clear information about historic glacier meltwater flows. Aside from the general observation, stream and spring discharges are lower now than they have been in the past.

Reference:

La Frenierre, J., 2014, "Asessing the Hydrologic Implications of Glacier Recession and the Potential for Water Resources Vulnerabilities in Volcán Chimborazo, Ecuador", PhD Dissertation, Ohio State University, 2014.

*P6 L10-12: Lack of any rainy/wet season samples is a limitation.*

We agree. We will acknowledge this shortcoming of the hydrochemical mixing model results more explicitly, as well as explain that the model simulations fill this gap to examine wet season in addition to dry season conditions. As we mention (p. 3, lines 6-7), mixing model analyses of melt contributions typically have been applied in the dry season in the better-studied outer tropics. We followed suit for a couple of reasons. First, melt contribution is often of greatest interest for water resource management during times of low precipitation. Second, dry season samples are easier to interpret in mixing models, because there is no need to consider precipitation as an additional end-member, which can consist of different hydrochemical signatures depending on the accumulation of solutes during runoff generation.

*Section 3.2.2: Having run same analytes in different labs in different years potentially introduces error or uncertainty. How confident are you in comparing different lab results? E.g. were detection limits the same, were any lab comparisons done?*

We had this same concern and had thus checked for consistency by comparing the concentrations of the different cations and anions across the different sampling periods. We found that the bulk concentrations (e.g., sum of cations in Figure 2(a) and (c)) at a certain location were similar, and that the spatial trends for each analyte were consistent across sampling periods - e.g., the concentrations generally increased moving downgradient in each sampling period. There were some systematic biases for certain analytes - e.g., February 2017 had $Cl^-$ concentrations at all locations that were higher by a relatively consistent difference compared to other sampling periods. However, this type of systematic bias between sampling periods is unimportant, because each implementation of the mixing model is carried out only with data within a certain sampling period, ensuring that we are not combining potentially incompatible data. We will explain this in the revised manuscript and include individual analyte plots over location for each sampling period in the Supplementary Information.

*Figure 2: 2(a) and 2(c) read like results.*

We agree, and in fact, Fig. 2(a) and (c) with concentration results are not discussed until the beginning of Section 3 Results and Discussion (Page 10, Line 31). We originally thought to combine these concentration results with mixing model configuration in Fig. 2(b) in Methods so that the reader can easily align the two. However, the reviewer's comment makes us realize that readers might prefer to see the concentration results in a separate figure in the Results section. We will accordingly move 2(a) and (c) to a separate figure than 2(b).

*P8 L20: grammar – 'is be unique'*

Thank you for catching this. It will be corrected in the manuscript.

*P10 L10: how were T, P, and RH interpolated?*

We used temperature and precipitation lapse rates, as described earlier in Section 3.1. Relative humidity measured at the Boca Toma station was applied over the entire watershed, because we did not have measurements elsewhere. We had inadvertently omitted the explanation about relative humidity and will add it to the Methods section. Although relative humidity does vary in reality, model sensitivity tests showed discharge simulations to be far less sensitive to relative humidity inputs than precipitation and temperature. We will revise the line mentioned in this comment to remind the reader that the interpolation approach was provided in the Methods section.

*P11 L4-7: unclear here how you ultimately selected tracers for the mixing model. E.g. were thresholds applied to bivariate plots?*

We will clarify that we chose as tracers those analytes that visually showed the mixed sample falling on a line between its two source samples based on the bivariate plots in Figures S3-S6. For example, for a tributary, the outflow sample should fall on a line between its two inflow sources. This comparison is done for all reaches and tributaries within a sampling period.

*P11 L13-17: Any hypothesis on why groundwater discharge was so low in Feb 2017? Are there temps or precip events that inform this anomaly?*

We intended to explain this in the text, but we now realize there was a typo. We meant to write: "However, the absolute contribution, determined by applying estimates of melt fractions to average observed weekly discharge measurements around the sampling time, was lowest in February 2017, because of significantly less total [NOT groundwater] discharge compared to the other sampling periods (Figure 3(b))." The lower total discharge was likely due to lower precipitation and temperature during the weeks around the sampling period compared to during the other sampling periods (Figure S1). We will edit the manuscript to correct the typo and to point the reader to Figure S1.

*Figure 4 caption, line 4: "corresponding to the"*

We will make this correction to the caption of Figure 4.

*P14 L13-15: how do these melt factors compare to the literature?*

They fall within the range of melt factors calculated for other glaciers in the tropics (3.5-9.9 mm we $^{\circ}C^{-1}d^{-1}$) reported in Fernandez and Mark (2016). We will add this.

Reference:

Fernandez and Mark, 2016, "Modeling modern glacier response to climate changes along the Andes Cordillera: A multiscale review". Journal of Advances in Modeling Earth Systems, DOI:10.1002/2015MS000482.

*P14 L24: reference for historic geodetic mass balance estimates?*

We will add the explanation that glacier volume change of debris-covered ice was estimated by differencing a GPS-validated photogrammetric digital elevation model in 1997 (Jordan et al., 2008) and terrestrial laser scanner (Riegl LMS-Z620) surveys in 2012 and 2013 (La Frenierre, 2016). We note that this estimate was not directly used in our

model simulations but instead only served as a comparison point for our calibrated melt estimate.

References:

Jordan, E., J. Gonzales, K. Castillo, J. Torres, L. Ungerechts, F. Velez, D. Blanco, and M. Cruz. 2008. “Ortofotomapa Del Chimborazo Y Su Valor Como Diagnóstico Para Cambios Climáticos En Relación Con Otros Glaciares Tropicales.” In Glaciares, nieves y hielos de América Latina. Cambio climático y amenazas, edited by Arenas, C. D. L., and Cadena, J.R., 239–60. Bogota, Columbia: Instituto Colombiano de Geología y Minería.

La Frenierre, J (2016), “Rapid downward deflation of a tropical-debris covered glacier: an analysis from Volcán Chimborazo, Ecuador”. American Geophysical Union (AGU) Fall Meeting, San Francisco, December 2016.

*P14 L30: missing close parentheses - “full details).”*

We will correct this in the manuscript.

*Figure 5 caption: clarification on “(a) average air temp below ELA (5050 m.a.s.l.) and over glaciers and simulated melt inputs” – does this mean T is averaged over ablation zone plus snow covered area?*

Yes. We will edit the caption so that this is clearer.

*Figure 5 caption, L4: ‘distribution’ should be ‘contribution’*

The change will be applied to the caption. Thanks.

*P20 L10-11: what you suggest here is a buffer against lower extreme low flows during drought times, which somewhat contradicts your repeated assertion (e.g. P20 L14) that melt does not necessarily provide the buffer often credited to it. Reconciling these, perhaps with a clear acknowledgement in the conclusions that the buffer does exist for extreme low flow scenarios, but the modulating effect in other flow scenarios may not*

*be as strong of a control on streamflow as other studies suggest.*

Thank you, this is a very good point. You captured exactly what we meant. We will revise the manuscript to clarify our conclusion that the classic paradigm that melt buffers does not always apply, though it still can buffer against extreme low discharge periods.

*P20 L26-27: these longer periods controlled by melt inputs are via infiltration and groundwater recharge, right?*

Yes, that is right. We will revise the sentence to make this clearer.

*Figure 8 caption, lines 1-2: reference here to glacial meltwater is misleading, since what you've characterized is glacier outflow that is a combo of ice and snow melt, right?*

No, what we represented here is only the result of glacier meltwater, since the same snowmelt amount was present in both the "With melt" and "No melt" simulation scenarios. As stated in an above response, we realized that we should have called the "No melt" scenario "No glacier melt", because it only removed glacier melt. We will revise the text to specify "With glacier melt" and "No glacier melt", and clarify that both scenarios include the same amount of snowmelt.

*P22 L9: "Recharge by meltwater"*

Thanks. We will modify this in the manuscript.

*P22 L22-23: Unclear what justifies the assertion that discharge could be reduced by half. Equilibrium discharge with glacier melt contributions and equilibrium discharge post-glaciers should be the same if precip is the same, barring other changes (e.g. increased ET). The peak water period in the middle is a different story, but this claim seems unsupported.*

As mentioned in Section 2 Study Area, the glaciers on Chimborazo are already retreating fast and thus are not in equilibrium. This leads us to consider the glacial meltwater as originating largely from stored ice from earlier time periods. However, the reviewer

makes a good point that even during peak water period (when ablation > accumulation), some of the meltwater could still originate from newly accumulated ice, such that post-glaciers, this amount of water would still fall as precipitation and contribute to discharge. We will edit our manuscript to clarify that without glaciers, but assuming the same precipitation, the potential future reduction of discharge by half is an upper limit, and that the reduction could be less if some of current-day precipitation goes to glacier accumulation. Similarly, we will acknowledge that the reduction could be less if the estimated current meltwater contribution includes snowmelt. Although our simulated scenarios aim to isolate the contributions of glacier melt and snowmelt (see our response to the preceding comment), the exact partitioning is not well-constrained by observations; further, the mixing model estimate of meltwater contribution may include snowmelt and/or melt of freshly accumulated ice (see our response to overarching issue #2).

*P22 L24-25 Related to the previous comment and as mentioned at the beginning, the other huge caveat is that you are looking at a point 2km from a glacier terminus, so results absolutely cannot be implied to inform understanding of vulnerability of water resources. Extrapolating further downstream is a logical next step and I think expanding your methods downstream would be an incredibly valuable contribution to this understanding!!!*

We agree that it would be extremely valuable to extend our work downstream, now that we have established this multi-method approach. As described in our response to the reviewer's overarching issue #1, we did apply the mixing model to the irrigation diversion point Boca Toma downstream from the Gavilan Machay discharge point. The findings and implications are provided in that first response. We were not able to extend the model over the entire Boca Toma watershed (26 $km^2$) due to the unavailability of weather input data for that other portion of the watershed outside of the Gavilan Machay watershed. Also, the other part does not have glacier melt, and so working on Gavilan Machay allows us to focus on the glacier contribution to the irrigation system.

---

## Author Response (AR1)

**Response to Reviewer 1**

We would like to thank the reviewer for their time to review our paper. In this response, we have addressed the reviewer's comments by providing clarifications and indicating how we edited the manuscript. The reviewer's comments are copied with a gray background, and our responses are provided with a white background.

*This paper presents a detailed multi-method assessment of glacier melt and groundwater contribution to runoff for a small catchment in the tropics. The authors find significant contributions of melting to overall runoff using tracer studies, time series analysis and hydrological modelling. They also show that melt water can be a substantial contributor to groundwater discharge. This is an excellent study that presents a thorough analysis of field data and modelling leading to interesting conclusions. The manuscript is very well written, and methods and results are clearly described. The findings are also nicely presented. Overall, I absolutely recommend this paper for publication in HESS after some mostly minor issues have been resolved (see below).*

We are encouraged by the reviewer's positive comments and have carefully addressed all issues raised.

*More substantive comments:*

*Page 5, line 11: Is there an estimate how important glacier-derived runoff is for the larger catchment? A high importance (irrigation system) is implied here, but how does the glacier runoff volume relate to larger-scale effective precipitation? Given that the absolute runoff amounts in the Gavilan Machay basin are really small (in the order of 0.1 $m^3/s$) I doubt that this water (despite of originating from the headwaters) has a major significance lower downstream. This is also supported by the statement of page 12, line 6. The glaciers' importance for water resources in the region might need to be better put into context.*

When extended downgradient to the Boca Toma diversion point, our mixing model analysis with HBCM predicts that surficicial runoff of meltwater contributes a range of 4-15% of the discharge to the irrigation system over 2012-2017, with the rest supplied by groundwater. While this melt contribution indeed seems to comprise a minor proportion, earlier investigation by La Frenierre (2014) on downstream water usage showed that farming communities cannot afford to lose any of the water; already, the irrigation system consistently fails to deliver its current allocations. Furthermore, if groundwater at Gavilan Machay also contains meltwater, as our simulations suggest, the actual total amount of meltwater contribution could be even higher than the 4-15% estimated for surficial runoff of meltwater. A lack of model input data outside of the Gavilan Machay sub-catchment prevented further extension of the model to Boca Toma. We added this discussion to **Page 24, line 28**.

*Page 9, line 23: The authors use a model that computes snow melt based on the energy balance. It is surprising to me that they nevertheless decided to implement an empirical, strongly simplified model for ice melt. This seems to be an unnecessary and also unphysical combination of approaches. Later, it is stated that a temperature index model is the only feasible approach given the limited data availability. However, if data are available to force an energy balance model for snow, it should also be applicable to glacier ice (just having a different albedo and surface roughness). More argumentation is required here, and possibly more insight into the energy balance scheme of Flux- PIHM.*

We do not use energy balance calculations for glacier melt for two reasons. First, energy balance calculations of glacier melt would have to be coupled with the other energy balance calculations

already in the Flux-PIHM model (for snow-melt, ET, sensible heat flux, and ground heat flux) because of both its role in the partitioning of incoming net radiation and its effect on surface temperature. However, adding this to Flux-PIHM requires intensive source-code modifications that are beyond the scope of this study. Second, an alternative approach of an approximate, uncoupled energy balance calculation of glacier-melt would be complicated by the lack of radiation input measurements in the study watershed. We currently use GLDAS data with Flux-PIHM for its energy balance calculations, but there is substantial uncertainty in applying the coarse-scale GLDAS radiation values over the steep mountainous watershed. Because of these difficulties, we chose to invoke the simpler temperature-index model and focused on constraining glacier-melt amounts based on discharge observations at the watershed outlet. We added this explanation on **page 10, line 10**.

We do note that using coarse-scale GLDAS does introduce uncertainty into the current Flux-PIHM energy balance calculations, including for snowmelt. However, even without partitioning some of the incoming radiation for glacier melt in the model, our simulated snowmelt is a relatively small contribution of the total meltwater (15%), suggesting that precipitation limitations may make snowmelt calculations less sensitive to uncertainties in radiation inputs. This is now explained on **page 16, line 5**.

*Page 9, Eq. 1: Given that relatively large parts (those experiencing the highest melt rates) of the glaciers are covered by supraglacial debris, I wonder how the model distinguishes between ice melting over these regions in comparison to clean ice.*

We had reported measurements of a slower ablation rate (0.54 to 0.87 m/yr) for the insulated debris-covered ice compared to a faster rate (3.4 m/yr) for the clean ice (**page 16, line 1**), which indeed support debris-dependent melt conditions. However, these were only a handful of ablation measurements over different time periods, which were not sufficient to constrain separate melt factors for debris-covered and clean ice. We state this more explicitly now on **page 15, line 33**.

*Figure 4: In my print-out (but not in the online pdf version!) there are ugly black squares around panels c and d, mostly covering the axis text. Please carefully check the figure data. Obviously, these issues only arise for particular printer drivers but make the figure almost unreadable. The same observation has also been made for Figure 8 (black squares left of the glacier snout in all panels).*

We have edited the figure (now Figure 5) and hope this resolves the issue.

*Page 20, line 12: Tackling the problem using different complementary approaches is highly beneficial. However, after reading the results section I somewhat missed a synthesis (figure) of the findings from the three different methodologies. For example, Fig. 3 and Fig 5 c/d could be combined to permit a direct comparison of findings based on tracers and based on the hydrological model which might also be helpful in discussing drawbacks and potentials of the individual methods.*

Thank you for the suggestion. We updated Figure 6d (formerly Figure 5d) (showing model results) by shading the interval of % Melt Contribution estimated with the mixing model (from Fig. 3a) in order to facilitate comparisons between methods. We point to this shading in the text (**page 20, line 7**). The reviewer's other suggestion of adding the discharge estimates from the mixing model (from Figure 3b) to Figure 6c would likely make the plot too busy, since it already has 5 different lines. The discharge information is summarized in Figure 6d, so we think that adding the mixing model results to Figure 6d should suffice to address the reviewer's concern.

*Additional detailed comments:*

*Page 2, line 11: normally, references are ordered with the year of appearance but not here.*

Thank you. We corrected this.

*Page 3, line 5: please shortly mention the physical reason (energy balance) why higher humidity leads to more ablation  this might not be immediately clear to the reader.*

We added the following explanation to **page 3, line 6**: "Harpold and Brooks (2018) showed that increasing humidity enhances ablation rate by increasing net longwave radiation and condensation."

*Page 5, line 19: Are there observations of recent glacier retreat in this region? Just to round up the story.*

We revised the following sentences on **page 5, line 1** by adding the underlined explanation: "Records since 1980 indicate that, consistent with the rest of the tropical Andes, temperatures have warmed $0.11°C$/decade around Volcń Chimborazo (Vuille et al., 2008; La Frenierre and Mark, 2017). This likely caused a 21% reduction in ice surface area and 180m increase in the mean minimum elevation of clean ice between 1986 and 2013 (La Frenierre and Mark, 2017)".

*Page 5, line 27: precipitation gradients were determined with stations at 3900 and 4500 m a.s.l., respectively. Will this elevation difference be enough to capture / estimate precipitation over the higher reaches of Chimborazo, i.e. between 5000 to 6200 m a.s.l.?*

Previous research in a glacierized mountainous watershed by Wang et al. (2016) found that the elevation-precipitation relationship is piecewise linear, with precipitation increasing with altitude below the elevation of maximum precipitation (EMP) and decreasing with altitude above the EMP. Such results support our application of a negative linear lapse rate calculated from our two stations - both located in the lower part of the watershed - to the higher elevation portions of the watershed. However, we should have and now do explicitly acknowledge that this assumes the EMP to be located below our watershed, which could lead to errors in the precipitation if the EMP is actually within the watershed above the lowest weather station. We point to the need for denser monitoring to better constrain the EMP and precipitation lapse rate, as well as recognize potential errors in the precipitation measurements due to wind and freezing temperatures. The assumptions and uncertainties about the precipitation spatial extrapolation approach are now discussed on **page 5, line 29**.

*Page 5, line 28: It is a drawback for melt model validation that the ablation stakes are only installed over a very limited elevation range (i.e. not permitting to capture elevation gradients in glacier melt), and  as it seems  only over the debris-covered parts of the glaciers. This should be stated.*

The ablation stakes were installed in clean ice. This had been explained previously in the Results section, but we have now also added it to the Methods section on **page 6, line 12**, where it should have been first mentioned. We also now better organized the presentation of the glacier melt measurement methods so that we explain in the next sentence (**page 6, line 16**) that we made imagery-based estimates of glacier mass loss of debris-covered ice. Later in Section 4.3.1 Calibration Results, we explain that our calibrated average glacier melt rate (below the equilibrium line altitude) is lower than the ablation stake measurements in faster-melting clean ice and higher than the mass balance measurements for the slower-melting debris-covered ice (**page 15, line 33**). We believe our reorganized discussion should now make this all clear.

*Figure 7: I like the analysis of the coherences and it allows interesting conclusions to be drawn. However, it would be helpful if the term "coherence" would be better introduced, making it clearer how it was computed and what it potentially shows.*

We present results for magnitude squared coherence, which can be thought of as the square of the correlation (between 0 and 1) between two variables at a certain frequency. Thus, coherencies between precipitation and discharge and between temperature and discharge indicate how strongly each of the climatic signals relate to discharge at a certain time scale. Looking at different time scales helps to distinguish whether these relationships may occur through fast surficial processes or slower subsurface processes, and whether discharge is more sensitive to certain climate forcings at particular time scales, represented by time frequencies. Mean squared coherence at time-frequency $f$ is defined as:

$$C_{xy} = \frac{|S_{xy}(f)|^2}{S_{xx}(f)S_{yy}(f)} \tag{1}$$

where, $S_{xx}(f)$ and $S_{yy}(f)$ are auto-spectral densities of variables $x$ and $y$, respectively, and $S_{xy}(f)$ is the cross spectral density of $x$ and $y$. We added this explanation of coherence to the Methods section on **page 6, line 26**.

*Page 19, line 12: Highly interesting finding. In how far could these 18% meltwater contribution to groundwater runoff be generalized to other catchments (different sizes, geology etc.)? Have there been other studies coming up with similar estimates or is this the first time this has been quantified? May be something for the conclusion section.*

Past studies have examined the overall role of groundwater in glacierized watersheds and have found it to contribute up to 80% of total stream discharge (Clow et al., 2003; Liu et al., 2004; Huth et al., 2004; Hood et al., 2006; Baraer et al., 2009; Andermann et al., 2012; Baraer et al., 2015; Pohl et al., 2015; Somers et al., 2016; Harrington et al., 2018). A smaller number of studies have also identified a component of meltwater in the groundwater (Favier et al., 2008; Lowry et al., 2010; Minaya, 2016; Baraer et al., 2015; Harrington et al., 2018), but to our knowledge, our work is the first to quantify this component. Generalizing our results to other glacierized watersheds depends on a number of geologic and climatic factors. The importance of meltwater contributions to stream-flow through groundwater depends first on the presence of groundwater pathways. These typically are most prominent with the presence of fractures in young volcanic bedrock (Tague et al., 2008; Frisbee et al., 2011; 2014; Markovich et al., 2016) - like Chimborazo - and sometimes even crystalline bedrock (Tague et al., 2009; Andermann et al., 2012; Pohl et al., 2015). However, morainic deposits (Favier et al., 2008, Minaya, 2016, Somers et al., 2016) and alpine meadow soils (Loheide et al., 2009; Lowry et al., 2010; Gordon et al., 2015) have also proved to be effective groundwater units below glaciers and snowpacks. Even in settings that may have limited groundwater networks extending throughout the watershed, talus slopes and rock glaciers can serve as localized areas of meltwater recharge (Clow et al., 2003; Baraer et al. 2015; Harrington et al., 2018). In the groundwater, the proportion of precipitation versus meltwater depends on watershed size and climate. Well-established discharge-watershed area relationships for non-glacierized watersheds lead to predictions of increased precipitation contribution in larger watersheds (with similar glacierized areas). More arid settings may be expected to have a higher proportion of glacier-melt due to overall less precipitation inputs to the watershed, although our results indicate a possible interaction between glacial melt contributions and precipitation, where rainfall boosts melt contributions through both the transfer of heat to glaciers and through antecedent moisture conditions that facilitate meltwater recharge. We thank the reviewer for this comment, which prompted us to better highlight our new contribution and its potential implications elsewhere. We have added a paragraph about this in the Conclusions on **page 24, line 1**.

*Page 22, line 22: I do not agree that runoff after glacier disappearance decreases by the current amount of melt contribution. As much as I understand, melt computed by the model includes both ice and snow melt. Whereas glacier ice melt is zero after the glacier has disappeared, snow melt is likely to remain a significant component of runoff or would be replaced by liquid precipitation in the case that the zero degree line remains above the top of Chimborazo all the time. Therefore, I would expect a significantly smaller runoff reduction for the catchment in the far future than implied here.*

The reviewer's comment prompts us to make one clarification and also qualify our statement about the runoff change after the glaciers disappear. First, we clarify that the model scenario we called "No Melt" should have been called "No glacier melt", and the scenario we called "With melt" should have been called "With glacier melt" - we did correct this naming scheme in the manuscript. Both model scenarios include the same snowmelt amount, because they use the same meteorological inputs to Flux-PIHM. Flux-PIHM simulates snowmelt based on precipitation and temperature inputs, while glacier melt is simulated externally (using the temperature-index model) and then added as another water source to Flux-PIHM. Thus, our calculation of change between the two scenarios isolates the effect of having glacier melt versus no glacier melt. This is now all clarified on **page 18, line 6**. Also, we have specified "glacier melt" rather than just generic "meltwater" in all the discussion about the simulation results.

Although by design our simulation scenarios aim to separate out glacier-melt and snowmelt contributions, we do lack constraints on their individual amounts, which means that our calibrated glacier-melt contribution could incorporate precipitation-sourced meltwater not fully accounted for in Flux-PIHM's snowmelt scheme - we now acknowledge this on **page 16, line 11** and **page 18, line 12**. Further, mixing model estimates of meltwater contribution uses meltwater from the glacier tongue, which may indeed include snowmelt and melt of freshly accumulated ice - this was already mentioned previously in the modeling results section, but we now introduce it more clearly in the Field Sampling section on p. 7, line 15. Thus, we now qualify our statement in the conclusion to say that the decrease in future discharge post-glaciers (under the same precipitation conditions) may be up to 50%, and that reduction could be less depending on snowmelt (**page 25, line 5**).

**Response to Reviewer 2**

We would like to thank the reviewer for their time to review our paper. In this response, we have addressed the reviewer's comments by providing clarifications and indicating how we edited the manuscript. The reviewer's comments are copied with a gray background, and our responses are provided with a white background.

*This paper is an excellent, in-depth exploration of multiple methods to constrain the role of meltwater in downstream hydrology, granted at a very small scale. The innovative contribution is the use of different time scales to demonstrate the interplay between melt regimes, precipitation patterns, and groundwater dynamics. There is tremendous opportunity to expand the relevance of this work in the future by applying a similar suite of methods to data from further downstream, or nested catchments.*

We are encouraged by the reviewer's positive comments and have carefully addressed all issues raised.

*A few overarching issues that should be more clearly addressed in the discus- sion/conclusions:*

*1. The 7.5 km$^2$ basin study area has offered valuable insights because of the data collection and monitoring that can be done at this scale, but it is important to acknowledge how your insights and results may translate downstream, given that your interpretations of the hydrology and implications of glacier recessions on discharge are being presented for an area in immediate proximity to the glacier terminus.*

We added a discussion of the potential downstream implications of our work through two new points in the manuscript.

First, on **page 24, line 28**, we now explain that we did extend our mixing model analysis downstream of Gavilan Machay to the Boca Toma diversion point for an irrigation system. We found that the surficial glacial meltwater contribution was about 4-15% of the discharge at Boca Toma. Although this amount of meltwater appears to comprise a minor proportion of discharge, an earlier investigation by La Frenierre (2014) on downstream water usage showed that farming communities cannot afford to lose any of the water. Already, the irrigation system consistently fails to deliver its current full allocations. Furthermore, if groundwater at Gavilan Machay contains glacial meltwater, as our simulations suggest, the actual total amount of meltwater contribution could be even higher than the 4-15% estimated for surficial meltwater. A lack of model input data outside of the Gavilan Machay sub-catchment prevented further extension of the model to Boca Toma.

Second, on **page 24, line 34**, we now discuss the potential outcome of extending measurements and the model implementation downstream. Looking within our small study watershed, estimates of groundwater contribution over the stream network reveal a nonlinear relationship with subcatchment area that contains sharp increases where geologic features likely create localized discharge points (we have added the figure to the Supplementary Information). This indicates that extrapolations downstream will likely depend on geological conditions that control groundwater, in addition to watershed size and climate inputs.

*2. The differentiation, or lack thereof, between snow and glacier melt should be more explicitly discussed. How big a role does snow (melt) play in the catchment, and what data to you have that informs this? To explore this, and relevant to many of your interpretations, a cursory estimate of the precipitation partition in the catchment could be interesting - what percent of precip falls as snow vs. rain based on your temperature and precip data? Given that information and your*

*discharge measurements, do you have a sense of relative contribution of snowmelt vs. glacier ice melt, or even how much of the discharge from the glacier terminus could also be liquid precip routed through that pour point?*

We have edited the manuscript to clarify the distinction between snowmelt and glacier melt in both the mixing model analysis and watershed model simulations.

For the mixing model, the meltwater end-member was represented using the hydrochemistry of a water sample collected just below the glacier tongue - this is now clarified on **page 7, line 15**. Therefore, our estimates of meltwater contribution using the mixing model approach would include both glacier and snowmelt.

In our watershed model simulations, we aimed to distinguish glacier from snowmelt, but we realized from the reviewer's comment that we did not explain this clearly in the text. When we referred to the "With melt" and "No melt" model scenarios, we meant "With glacier melt" and "No glacier melt", respectively, because the two scenarios actually have the same amount of snowmelt; the only difference is that the former has an added glacier melt amount of water - determined with the temperature index model calibrated to discharge measurements. Apart from this glacier melt amount, both scenarios include the same meteorological inputs and thus same snowmelt amount. Through this approach, the melt contribution that we determined by differencing the two scenarios should only represent glacier melt. We have now clarified this on **page 18, line 6**.

The reviewer does make the good suggestion that we should discuss the relative snowmelt and glacier melt amounts in the model. The model Flux-PIHM simulates the partitioning of precipitation inputs between rain and snow based on air temperature. Over the 2016 period, the model predicted that about 12% of precipitation falls as snow in the watershed– this is now stated on **page 16, line 6**. The calibrated glacier melt is 567% of simulated snowmelt amount (i.e., snowmelt is 15% of the total meltwater) (now explained on **page 16, line 5**). On average, snow melt contributes 8% of stream discharge, while glacial melt contributes 52% (now stated on **page 18, line 8**).

We acknowledge that although we aim to distinguish between glacier melt and snowmelt in the model, there is uncertainty in the partitioning due to a lack of separate data constraints. Our snowmelt simulation is highly sensitive to any errors in the lapse-rate based spatial extrapolation of precipitation and temperature in the watershed, and so it is possible that our calibrated glacier melt includes some amount of snowmelt that is not captured in the Flux-PIHM simulation - this is now acknowledged on **page 16, line 11** and **page 18, line 12**.

*Specific comments:*

*P3 L23-24: which 4 major river systems?*

The four major river systems are: the Río Mocha (NE flank), Río Colorado (NW flank), Río Guano (SE flank), and Río Chimborazo (SW flank). The names of the watersheds were added to **page 3, line 23**.

*P5 L5: Do you know if historically any other glaciers generated perennial surface discharge?*

There are no historical streamflow data for any of Chimborazo's glacierized watersheds that are close enough to the mountain to be able to discern a glacier melt signature. Discussions with local water users by did La Frenierre (2014) not yield clear information about historic glacier meltwater flows, aside from the general observation stream and spring discharges are lower now than they have been in the past. We note the lack of historical glacier melt data on **page 5, line 5**.

*P6 L10-12: Lack of any rainy/wet season samples is a limitation.*

As we mentioned, mixing model analyses of melt contributions typically have been applied in the dry season in the better-studied outer tropics (**page 3, line 7**). We followed suit, because melt contribution is often of greatest interest for water resource management during times of low precipitation - this is now explained on **page 7, line 7**. We also now explain explicitly that the model simulations fill this gap to examine wet season in addition to dry season conditions (**page 7, line 8**).

*Section 3.2.2: Having run same analytes in different labs in different years potentially introduces error or uncertainty. How confident are you in comparing different lab re- sults? E.g. were detection limits the same, were any lab comparisons done?*

We had this same concern and had thus checked for consistency by comparing the concentrations of the different cations and anions across the different sampling periods. We found that the bulk concentrations (e.g., sum of cations in Figure 2(a) and (c)) at a certain location were similar, and that the spatial trends for each analyte were consistent across sampling periods - e.g., the concentrations generally increased moving downgradient in each sampling period. There were some systematic biases for certain analytes from a particular sampling period - e.g., February 2017 had $Cl^-$ concentrations at all locations that were higher by a relatively consistent difference compared to other sampling periods. However, this type of systematic bias between sampling periods is unimportant, because each implementation of the mixing model is carried out only with data within a certain sampling period, ensuring that we are not combining potentially incompatible data. We now explain this in the revised manuscript (**page 8, line 4**) and include individual analyte plots over location for each sampling period in the Supplementary Information (**Figure S3**).

*Figure 2: 2(a) and 2(c) read like results.*

We agree, and in fact, previous Figures 2(a) and (c) with concentration results are not discussed until the beginning of Section 3 Results and Discussion. We originally thought to combine these concentration results with mixing model configuration in previous Figure 2(b) in Methods so that the reader can easily align the two. However, the reviewer's comment makes us realize that read- ers might prefer to see the concentration results in a separate figure in the Results section. We accordingly moved previous Figures 2(a) and (c) to a separate figure (currently **Figure 3**).

*P8 L20: grammar 'is be unique'*

Thank you. We corrected this.

*P10 L10: how were T, P, and RH interpolated?*

We used temperature and precipitation lapse rates, as described earlier in the Methods section. Relative humidity measured at the Boca Toma station was applied over the entire watershed, because we did not have measurements elsewhere. We had inadvertently omitted the explanation about relative humidity previously and have now added it to the Methods section (**page 6, line 7**). The revised text points out that discharge simulations should be less sensitive to this approximate treatment of relative humidity compared to precipitation and temperature, which directly control water inputs to the watershed. This reviewer's comment on the Results section makes us realize that we should add a reference back to the Methods section on **page 10, line 31**.

*P11 L4-7: unclear here how you ultimately selected tracers for the mixing model. E.g. were thresholds applied to bivariate plots?*

We clarified on **page 11, line 22** that we chose as tracers those analytes that visually showed the mixed sample falling close to the line between its two source samples based on the bivariate plots in Figures S3-S6. For example, for a tributary, the outflow sample should fall on a line between its two inflow sources. This comparison is done for all reaches and tributaries within a sampling period.

*P11 L13-17: Any hypothesis on why groundwater discharge was so low in Feb 2017? Are there temps or precip events that inform this anomaly?*

We intended to explain this in the text, but we now realize there was a typo. We meant to write: "However, the absolute contribution, determined by applying estimates of melt fractions to average observed weekly discharge measurements around the sampling time, was lowest in February 2017, because of significantly less total [NOT groundwater] discharge compared to the other sampling periods (Figure 4(b))." The lower total discharge was likely due to lower precipitation and temperature during the weeks around the sampling period compared to during the other sampling periods (Figure S1). We edited the manuscript to correct the typo and pointed the reader to Figure S1 (**page 12, line 2**).

*Figure 4 caption, line 4: "corresponding to the"*

Thank you. We corrected this.

*P14 L13-15: how do these melt factors compare to the literature?*

They fall within the range of melt factors calculated for other glaciers in the tropics (3.5-9.9 mm we $°C^{-1}d^{-1}$) reported in Fernandez and Mark (2016). We added this on **page 15, line 23**.

*P14 L24: reference for historic geodetic mass balance estimates?*

We added the explanation that glacier volume change of debris-covered ice was estimated by differencing a GPS-validated photogrammetric digital elevation model in 1997 (Jordan et al., 2008) and terrestrial laser scanner (Riegl LMS-Z620) surveys in2012 and 2013 (La Frenierre, 2016) (**page 6, line 18**). In the same section, we also noted that due to imitations in the spatiotemporal coverage of the glacier mass balance measurements, this estimate was not directly used in our model simulations but instead only served as a comparison point for our calibrated melt estimate (**page 16, line 2**).

*P14 L30: missing close parentheses - "full details)."*

Thank you. We corrected this.

*Figure 5(d): y-axis label typo "Contribution"*

Thank you. We corrected this

*Figure 5 caption: clarification on "(a) average air temp below ELA (5050 m.a.s.l.) and over glaciers and simulated melt inputs" does this mean T is averaged over ablation zone plus snow covered area?*

We clarified the caption for **Figure 6a** (previously Figure 5a): the figure shows T averaged over the ablation zone. We do not separately track snow-covered area, as this is highly transient in the inner tropics.

*Figure 5 caption, L4: 'distribution' should be 'contribution'*

Thank you. We corrected this.

*P20 L10-11: what you suggest here is a buffer against lower extreme low flows during drought times, which somewhat contradicts your repeated assertion (e.g. P20 L14) that melt does not necessarily provide the buffer often credited to it. Reconciling these, perhaps with a clear acknowledgement in the conclusions that the buffer does exist for extreme low flow scenarios, but the modulating effect in other flow scenarios may not be as strong of a control on streamflow as other studies suggest.*

Thank you, this is a very good point. You captured exactly what we meant. We revised the text to clarify our conclusion that the classic paradigm that melt buffers does not always apply, though it still can buffer against extreme low discharge periods (**page 22, line 9**).

*P20 L26-27: these longer periods controlled by melt inputs are via infiltration and groundwater recharge, right?*

No, in fact we are talking about 30-80 day discharge patterns being driven by 30-80 day melt production patterns. We realize this is not an obvious connection, since we also talked about melt contributions to discharge being a fast (hourly time scale) process. But, this finding emerged from the coherence analysis (Figure 8). We clarified this in a few ways. First, we clarified the text where this connection was originally discussed with reference to coherence results, on **page 20, line 15**. Here, and through-out, we edited the wording to clarify connections between melt "production" (instead of the former wording of "melt input") versus melt contribution to discharge. In this particular line, we explicitly point to the coherence plot to show that there is a strong correlation at 30-80 day periods between glacier melt production and glacier melt contribution to discharge. In the following lines in that section, we more explicitly acknowledge that even though this is a dynamic at longer multi-month timescales, it is nonetheless occurring through surface runoff of meltwater. Second, we reworded the sentence in question in the Conclusion section so that it now reads: "Coherence analysis of the model results showed that not only were diurnal discharge patterns responding to radiation-driven melt inputs, but relative melt contribution and discharge variations over 30-80 day periods were controlled by extended glacier melt production periods that also contribute to discharge through surface runoff (Figure 9(c))." In particular, we reminded the reader that this came from the coherence analysis. We also emphasized that we are talking about longer periods of melt ("extended") and that we are talking about melt production and not vaguely melt "inputs."

*Figure 8 caption, lines 1-2: reference here to glacial meltwater is misleading, since what you've characterized is glacier outflow that is a combo of ice and snow melt, right?*

No, what we represented here is only the result of glacier meltwater, since the same snowmelt amount was present in both the "With melt" and "No melt" simulation scenarios. As stated in an above response, we realized that we should have called the "No melt" scenario "No glacier melt", because it only removed glacier melt. See our response above to your overarching issue #2 for how we clarified this.

*P22 L9: "Recharge by meltwater"*

Thank you. We corrected this.

*P22 L22-23: Unclear what justifies the assertion that discharge could be reduced by half. Equilibrium discharge with glacier melt contributions and equilibrium discharge post-glaciers should be the same if precip is the same, barring other changes (e.g. increased ET). The peak water period in the middle is a different story, but this claim seems unsupported.*

As mentioned in Section 2 Study Area, the glaciers on Chimborazo are already retreating fast and

thus are not in equilibrium. This leads us to consider the glacial meltwater as originating largely from stored ice from earlier time periods. In fact, our calibrated glacier melt input is addition to precipitation inputs, so we consider it to represent melt of pre-existing ice at the start of the model run - this is now explained on **page 18, line 10**. However, the reviewer makes a good point that even during peak water period (when ablation $\succ$ accumulation), some of the meltwater could still originate from newly accumulated ice, such that post-glaciers, this amount of water would still fall as precipitation and contribute to discharge. We edited our manuscript to clarify that without glaciers, but assuming the same precipitation, the potential future reduction of discharge by half is an upper limit, and that the reduction could be less if some of current-day precipitation goes to glacier accumulation, or if the estimated current meltwater contribution includes snowmelt (**page 25, line 5**).

*P22 L24-25 Related to the previous comment and as mentioned at the beginning, the other huge caveat is that you are looking at a point 2km from a glacier terminus, so results absolutely cannot be implied to inform understanding of vulnerability of water resources. Extrapolating further downstream is a logical next step and I think expanding your methods downstream would be an incredibly valuable contribution to this understanding!!!*

We agree that it would be extremely valuable to extend our work downstream, now that we have established this multi-method approach. As described in our response to the reviewers overarching issue #1, we did apply the mixing model to the irrigation diversion point Boca Toma downstream from the Gavilan Machay discharge point. The findings and implications are provided in that first response. We were not able to extend the model over the entire Boca Toma watershed (26 $km^2$) due to the unavailability of weather input data for that other portion of the watershed outside of the Gavilan Machay watershed. Also, the other part does not have glacier melt, and so working on Gavilan Machay allows us to focus on the glacier contribution to the irrigation system.

[revised manuscript text omitted]

**Figures**

[Figure]

**Figure S1.** Comparison of average weekly discharge ($m^3$/s) at Gavilan Machay and rainfall (mm/hr) at Boca Toma. Vertical black dashed lines indicate weeks where sampling occurred. Despite gaps in data, it can be seen that precipitation was higher in the time surrounding the 2015 sampling campaign than in the times surrounding the other campaigns.

[Figure]

**Figure S2.** The reach and confluences cells relative to a stream system.

[Figure]

**Figure S3.** The concentrations of anions and cations across the different sampling periods.

[Figure]

**Figure S3.** The concentrations of anions and cations across the different sampling periods.

[Figure]

**Figure S4.** Bivariate diagrams of tracers selected for January 2012 analysis. The solid black lines represent linear regressions through all samples and dashed black lines indicated the regressions' 95% confidence intervals. Samples from the main Rio Mocha channel before and after the Gavilan stream joins it (filled green circles) consistently plot outside or away from the mixing line created between groundwater and meltwater. MOCH-S7 was not considered in the analysis (shown with hallow square). These samples are also responsible for the poor R2 and p-values of sulfate and chloride.

**July 2012**

[Figure]

**Figure S5.** Bivariate diagrams of tracers selected for July 2012 analysis. The solid black lines represent linear regressions through all samples and dashed black lines indicated the regressions' 95% confidence intervals. Samples from the main Rio Mocha channel before and after the Gavilan stream joins it (filled green circles) consistently plot outside or away from the mixing line created between groundwater and meltwater. MOCH-S7 was not considered in the analysis (shown with hollow square). These samples are also responsible for the poor R2 and p-values of sulfate and chloride.

**June 2015**

[Figure]

**Figure S6.** Bivariate diagrams of tracers selected for June 2015 analysis. The solid black lines represent linear regressions through all samples and dashed black lines indicated the regressions' 95% confidence intervals. Samples from the main Rio Mocha channel before and after the Gavilan stream joins it (filled green circles) consistently plot outside or away from the mixing line created between groundwater and meltwater. MOCH-S7 was not considered in the analysis (shown with hallow square). These samples are also responsible for the poor R2 and p-values of sulfate and chloride.

**June 2016**

[Figure]

**Figure S7.** Bivariate diagrams of tracers selected for June 2016 analysis. The solid black lines represent linear regressions through all samples and dashed black lines indicated the regressions' 95% confidence intervals. Samples from the main Rio Mocha channel before and after the Gavilan stream joins it (filled green circles) consistently plot outside or away from the mixing line created between groundwater and meltwater. MOCH-S7 was not considered in the analysis (shown with hallow square). These samples are also responsible for the poor R2 and p-values of sulfate and chloride.

[Figure]

**Figure S8.** Bivariate diagrams of tracers selected for February 2017 analysis. The solid black lines represent linear regressions through all samples and dashed black lines indicated the regressions' 95% confidence intervals. Samples from the main Rio Mocha channel before and after the Gavilan stream joins it (filled green circles) consistently plot outside or away from the mixing line created between groundwater and meltwater. MOCH-S7 was not considered in the analysis (shown with hallow square). These samples are also responsible for the poor R2 and p-values of sulfate and chloride.

[Figure]

**Figure S9.** Hierarchical cluster analysis dendrograms for tracers used in HCBM for January 2012, July 2012, June 2015, June 2016, and February 2017. Sample grouping verifies effectiveness of selected tracers in distinguishing between source waters, as groundwater samples (GW) cluster separately from melt water (Melt) samples. GW-7 is considered an outlier, and is suspected to be a mixed source sample or to originate from a unique geology.

[Figure]

**Figure S10.** Power spectral density of **(a)**discharge and temperature, **(b)** discharge and precipitation

[Figure]

**Figure S11.** Cross-correlation of **(a)**hourly discharge and precipitation, **(b)** weekly discharge and temperature

[Figure]

**Figure S12.** Calculated absolute groundwater discharge versus percentage of total watershed area drained for Gavilan Machay subcatchment (McLaughlin, 2017).

**Hydrochemical Basin Characterization Model (HBCM)**

The following set of mass-balance equations for J tracers applies to each HBCM cell:

$$C_{tot_j} = \frac{\sum_{i=1}^{I}(C_{i,j} * Q_i) + \epsilon_j}{Q_{tot}} \tag{S1}$$

Where:

$j$: index for a specific natural tracer, 1 through $J$

$i$: index for a specific source to the cell (ice, tributary, or groundwater), 1 through $I$

$C_{tot_j}$: concentration of tracer $j$ at the cell outlet

$C_{i,j}$: concentration of tracer $j$ in source $i$

$Q_{tot}$: Total discharge at cell outlet

$Q_i$: Contribution to discharge from source $i$

$\epsilon_j$: residual error between the observed and predicted concentration flux out of the cell for tracer $j$

HBCM solves for the unknown relative contributions of each source ($Q_i/Q_{tot}$) by minimizing the sum of the residual errors:

$$\sum_{j=1}^{J} \epsilon_j \tag{S2}$$

In order to over-constrain the problem, the model requires $J = I$ tracers and, preferably, $J > I$ tracers should be utilized in order to avoid the possibility of correlated tracers that do not independently constrain the problem. HBCM checks that a tracer is conservative within each cell via three tests:

1. A tracer value in a cell outflow cannot be outside the range bracketed by the possible contributors;

2. The tracer value at the cell outflow, along with that of at least one input component, must be greater than the detection limit of the analytical methods (confirmed by user); and

3. There must be a minimum 5% difference between the concentration of a tracer from each source.

If any of these requirements is not met, HBCM will reject the tracer for use in the cell.

**Tables**

**Table S1.** HBCM analysis result for January 2012 to February 2017 with different cell configurations. – See the spreadsheet "HBCM Table".

| | Observed | | | | Flux-PIHM pedotransfer function results | | | | | | |
|---|---|---|---|---|---|---|---|---|---|---|---|
| | Sand (%mass) | Silt (%mass) | Clay (%mass) | OC (%mass) | KINF (m/s) | KSATV (m/s) | KSATH (m/s) | SMCMAX (-) | SMCMIN (-) | ALPHA (1/m) | BETA (-) |
| Podwojewski et al. 2002, Pantano 60-80 | 26 | 41 | 8 | 12 | 4.52E-07 | 6.44E-07 | 6.44E-06 | 0.479 | 0.05 | 1.908 | 1.157 |
| Patano 80-95+ | 33 | 39 | 14 | 8 | 1.47E-06 | 1.58E-06 | 1.58E-05 | 0.481 | 0.05 | 3.491 | 1.136 |
| Humid páramo 0-15 | 26 | 44 | 7 | 10 | 6.24E-07 | 1.03E-06 | 1.03E-05 | 0.48 | 0.05 | 2.267 | 1.173 |
| Humid páramo 15-30 | 32 | 41 | 8 | 7 | 1.51E-05 | 6.76E-06 | 3.05E-05 | 0.418 | 0.05 | 5.34 | 1.26 |
| Dry páramo 0-15 | 30 | 43 | 8 | 7 | 8.28E-05 | 3.71E-05 | 6.96E-05 | 0.493 | 0.05 | 5.82 | 1.22 |
| Dry páramo 15-30 | 35 | 33 | 20 | 5 | 8.28E-05 | 3.71E-05 | 6.96E-05 | 0.493 | 0.05 | 5.82 | 1.22 |
| Minaya (2016), Low Elev. | 20.28 | 31.98 | 6.24 | 22 | 3.71E-08 | 8.42E-08 | 8.42E-07 | 0.482 | 0.05 | 0.327 | 1.144 |
| Minaya (2016), Middle Elev. | 26.07 | 30.81 | 11.06 | 21 | 6.13E-08 | 1.03E-07 | 1.03E-06 | 0.479 | 0.05 | 0.466 | 1.124 |
| Minaya (2016), Highest Elev. | 23.4 | 39.6 | 6.3 | 10 | 4.28E-07 | 1.70E-07 | 1.70E-06 | 0.465 | 0.05 | 1.442 | 1.1 |

**Table S2.** Páramo soil measurments applied into pedo transfer functions (Podwojewski et al., 2002; Minaya, 2016)